# Bacterial Community Shifts during Polyp Bail-Out Induction in *Pocillopora* Corals

Po-Shun Chuang,[a]* Yosuke Yamada,[a]§ Po-Yu Liu,[b]∞ Sen-Lin Tang,[b] Satoshi Mitarai[a]

[a]Marine Biophysics Unit, Okinawa Institute of Science and Technology Graduate University, Okinawa, Japan
[b]Biodiversity Research Center, Academia Sinica, Taipei, Taiwan (ROC)

**ABSTRACT** Polyp bail-out constitutes both a stress response and an asexual reproductive strategy that potentially facilitates dispersal of some scleractinian corals, including several dominant reef-building taxa in the family Pocilloporidae. Recent studies have proposed that microorganisms may be involved in onset and progression of polyp bail-out. However, changes in the coral microbiome during polyp bail-out have not been investigated. In this study, we induced polyp bail-out in *Pocillopora* corals using hypersaline and hyperthermal methods. Bacterial community dynamics during bail-out induction were examined using the V5-V6 region of the 16S-rRNA gene. From 70 16S-rRNA gene libraries constructed from coral tissues, 1,980 OTUs were identified. *Gammaproteobacteria* and *Alphaproteobacteria* consistently constituted the dominant bacterial taxa in all coral tissue samples. Onset of polyp bail-out was characterized by increased relative abundance of *Alphaproteobacteria* and decreased abundance of *Gammaproteobacteria* in both induction experiments, with the shift being more prominent in response to elevated temperature than to elevated salinity. Four OTUs, affiliated with *Thalassospira*, *Marisediminitalea*, *Rhodobacteraceae*, and *Myxococcales*, showed concurrent abundance increases at the onset of polyp bail-out in both experiments, suggesting potential microbial causes of this coral stress response.

**IMPORTANCE** Polyp bail-out represents both a stress response and an asexual reproductive strategy with significant implications for reshaping tropical coral reefs in response to global climate change. Although earlier studies have suggested that coral-associated microbiomes likely contribute to initiation of polyp bail-out in scleractinian corals, there have been no studies of coral microbiome shifts during polyp bail-out. In this study, we present the first investigation of changes in bacterial symbionts during two experiments in which polyp bail-out was induced by different environmental stressors. These results provide a background of coral microbiome dynamics during polyp bail-out development. Increases in abundance of *Thalassospira*, *Marisediminitalea*, *Rhodobacteraceae*, and *Myxococcales* that occurred in both experiments suggest that these bacteria are potential microbial causes of polyp bail-out, shedding light on the proximal triggering mechanism of this coral stress response.

**KEYWORDS** polyp bail-out, *Pocillopora*, microbiome, bacterial community, *Thalassospira*

Address correspondence to Po-Shun Chuang, ps.bob.chuang@gmail.com.

*Present address: Po-Shun Chuang, Biodiversity Research Center, Academia Sinica, Taipei, Taiwan (ROC).

§Present address: Yosuke Yamada, Institute for Extra-cutting-edge Science and Technology Avant-garde Research, Japan Agency for Marine-Earth Science and Technology, Kochi, Japan.

∞Present address: Po-Yu Liu, School of Medicine, College of Medicine, National Sun Yat-sen University, Kaohsiung, Taiwan (ROC).

The authors declare no conflict of interest.

Contemporary climate change has inflicted great stresses on tropical coral reefs. Ocean warming is pushing many tropical corals beyond their tolerance limits, causing more frequent massive bleaching events, including four of pan-tropical scale in 1998, 2010, 2015 to 2016, and 2019 to 2020 (1–4). Coral bleaching refers to a loss of endosymbiotic dinoflagellates (family Symbiodiniaceae) in coral hosts, which results in "whitening" (5). As most scleractinian corals utilize photosynthates from symbiotic dinoflagellates as their main carbon source, severe bleaching often results in mass coral mortality, due to energy deficiency and/or increased vulnerability to other stresses (6–9).

In addition to ocean warming, increased extreme weather events, e.g., precipitation and drought, due to alteration of atmospheric circulation patterns, can cause dramatic

fluctuations in seawater salinity (10, 11). As corals are osmoconformers with a limited range of salinity tolerance, salinity changes can impose additional stresses on tropical corals. Hypersaline seawater induces cellular stresses and reduces photosynthesis and/or respiration in stony corals (12–14). Interestingly, some scleractinian corals, e.g., *Siderastrea siderea* and *Fungia granulosa*, are able to acclimate to long-term hypersaline environments, possibly by modulating the associated microbiome (13, 14). The ecological significance of this acclimation in corals, however, is still unclear, due to the scarcity of studies.

Coral polyp bail-out represents a stress response involving coenosarc tissue degradation and detachment of solitary polyps (15). As bailed-out polyps can resettle once stresses are relieved, this phenomenon has been proposed as an asexual reproduction strategy (16–18). Although few data are available regarding polyp bail-out in the field (15, 19), laboratory experiments have demonstrated induction of this response in several coral species under various conditions, including treatments with hypersaline or hyperthermal seawater (18, 20). Unlike bleaching, polyp bail-out generates free-living, uncalcified polyps that can drift away in currents, allowing corals to escape local stresses. Before resettlement, detached polyps can survive for weeks to months (18, 20–22), enabling them to disperse much farther than by traditional reproductive methods, such as spawning, brooding, and fragmentation. Indeed, polyp bail-out has been proposed as a possible invasion strategy of the orange cup coral, *Tubastraea coccinea* (23, 24). This stress response therefore may serve to preserve tropical scleractinian corals as well as to reshape coral reef distributions in the face of global climate change.

Coral-associated bacteria (CAB) constitute an important part of coral holobionts, contributing to coral physiology and pathology and determining coral health (25, 26). Alteration of associated bacterial communities is a common coral response to environmental fluctuations, which allows rapid acquisition of advantageous coral symbionts in changed environments (27–30). However, reorganization of CAB also enables invasion of opportunistic pathogens (28). Severe or prolonged stresses may also cause irreversible microbiome changes, which in turn reduce fitness of coral holobionts (25). Recently, genetic studies on polyp bail-out have pointed to a possible link between coral microbiomes and onset of this stress response. Asynchronous expression of tumor necrosis factor (TNF) and tumor necrosis factor receptor (TNFR) genes in *Pocillopora* corals during bail-out induction led to a hypothesis of a microbial trigger for initiation of polyp bail-out (31, 32). Activation of the Toll-like receptor signaling pathway and immune responses also implies changes of coral-microbiome interactions during polyp bail-out progression (33). However, to our knowledge no study has so far attempted to characterize microbes responsible for initiation of polyp bail-out.

In the present study, we provide the first CAB data collected during development of polyp bail-out, induced in *Pocillopora* corals by hypersaline and hyperthermal treatments. These results broaden our understanding of polyp bail-out and shed light on methods to promote polyp recovery and resettlement.

## RESULTS

**Coral polyp bail-out and species identification.** To investigate microbiome dynamics during polyp bail-out, fragments from five coral colonies were subjected to either a hypersaline or a hyperthermal treatment to induce polyp bail-out. In the hypersaline experiment, polyps retracted into corallites at 12 h in the treatment group in response to the stress (salinity at 43‰; Fig. 1). The hypersaline method induced polyp detachment in all coral samples in the treatment group at 24 h, when seawater salinity reached 47‰ (Fig. 1). For the hyperthermal experiment, polyp retraction was observed at day 2 (31℃). Unlike the hypersaline experiment, polyp detachment in the hyperthermal experiment was observed at two different times: at day 5 in two of the fragments (34℃; *Pocillopora* samples number 15 and number 24) and at day 7 in the remaining three fragments (34℃; *Pocillopora* samples number 10, number 22, and number 23; Fig. 1). As both represented onset of polyp detachment, samples at day 5 and day 7 were pooled (denoted days 5 to 7) in subsequent analyses (so too, corresponding samples in the control group). Detached polyps in both experiments

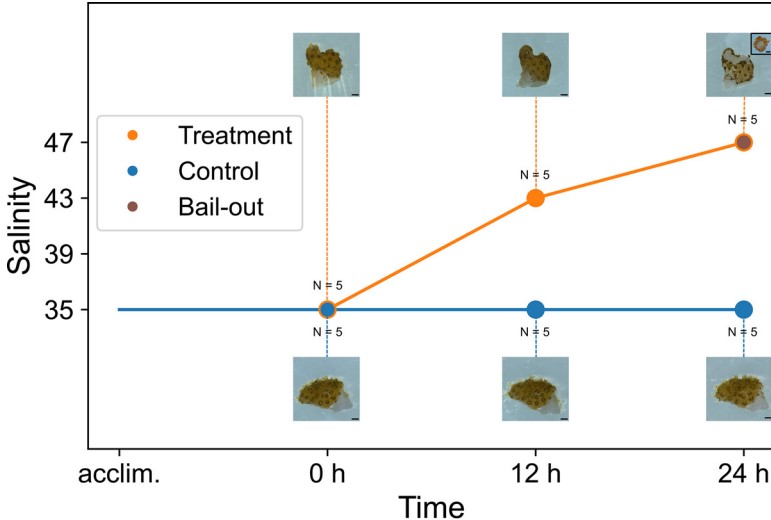

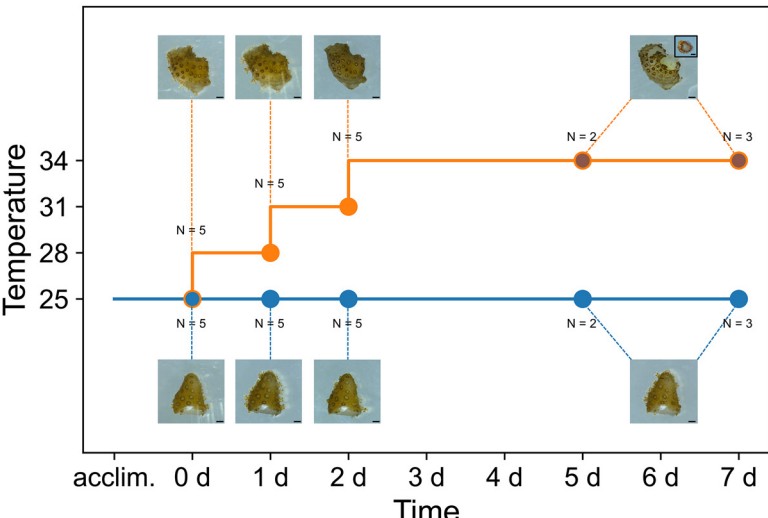

**FIG 1** Experimental designs of polyp bail-out induction using hypersaline (upper; salinity unit: ‰) and hyperthermal (lower; temperature unit: °C) strategies. Onset of polyp bail-out is highlighted with brown dots. Scale bar: 1 mm (200 $\mu$m in photos of bailed-out polyps).

showed intact morphological features similar to those reported in earlier studies (31, 33), including round to cylindrical bodies and intact tentacles, indicating successful induction of polyp bail-out instead of tissue sloughing or polyp death. No samples in the control groups showed any signs of bleaching, bail-out, or any other stress response in either experiment. Interestingly, one coral sample (*Pocillopora* number 15) was slightly bleached 1 day after incubation at 34°C (day 3) and was severely bleached by the time polyp bail-out was observed (Fig. S1). Genotyping based on phylogenetic analyses with published reference sequences (using neighbor-joining and maximum likelihood methods) grouped *Pocillopora* number 15 with *P. damicornis*, while the remaining 4 coral colonies employed in this study were grouped with *P. acuta* (Fig. S2).

**16S-rRNA data processing.** A total of 16,760,948 reads were generated from 72 16S-rRNA gene libraries, including 30 libraries from the hypersaline experiment (5 colonies * 2 conditions * 3 time points), 40 libraries from the hyperthermal experiment (5 colonies * 2 conditions * 4 time points), and 2 tissue-negative controls included at the DNA extraction step. Mothur successfully assembled 7,988,258 contigs, among which 247,181 nonredundant contigs were identified after aligning against the SILVA reference database. Deduplication and removal of chimeric and nonbacterial contigs further reduced this number to 35,159 unique contigs.

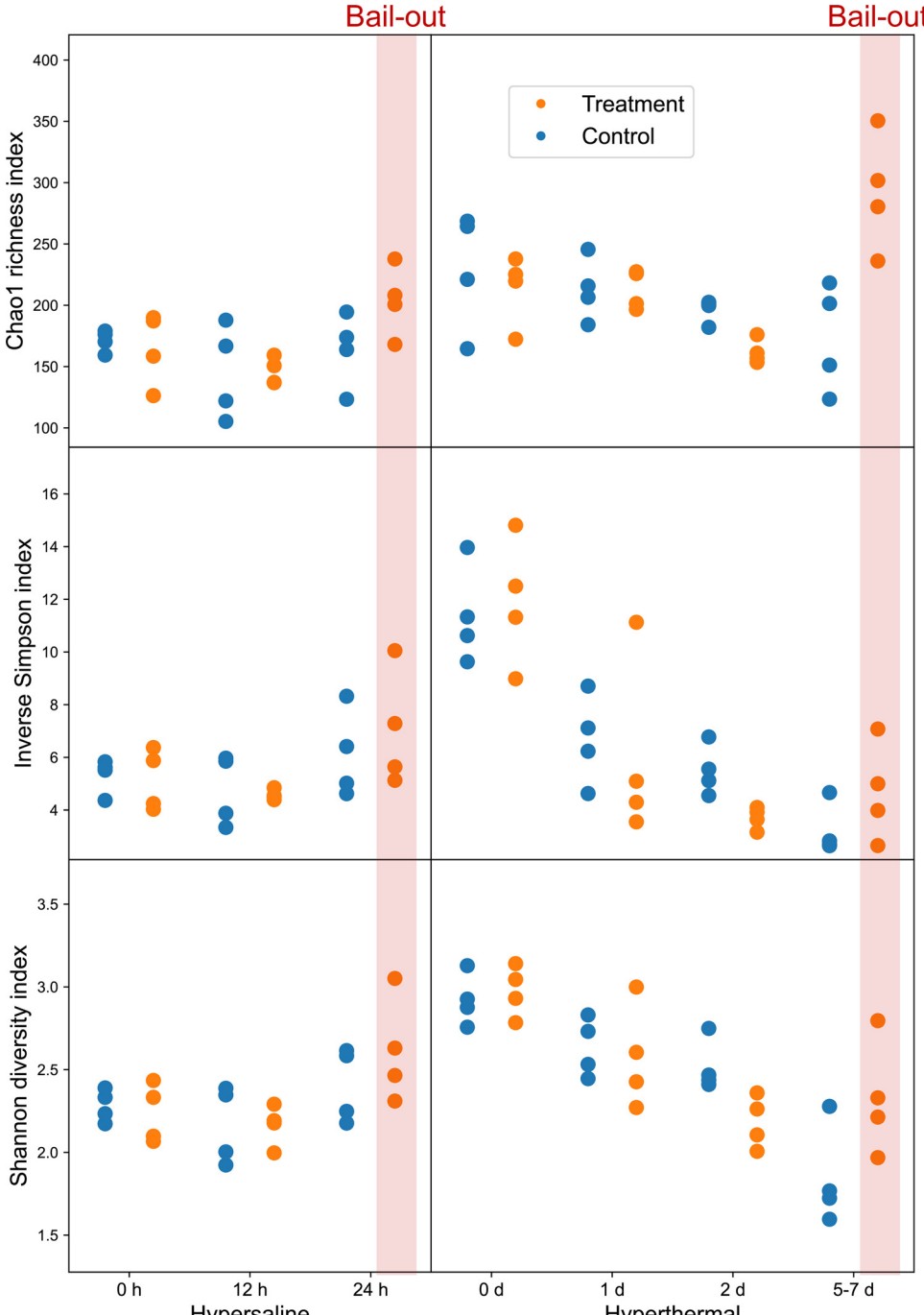

**FIG 2** Alpha diversity was estimated using Chao1 richness, inverse Simpson, and Shannon diversity indices. Onset of polyp bail-out is highlighted with red boxes.

After decontamination using microDecon, the remaining unique contigs (32,413) were classified into 1,980 bacterial Operational Taxonomic Units (OTUs), belonging to 52 bacterial classes. Sequencing depths were in the range of 70,259 to 137,568 sequences/library among constructed coral tissue libraries, with an average of about 103,291 sequences/library.

**Alpha diversity in constructed libraries.** Given that *Pocillopora* number 15 was identified as *P. damicornis* and showed discernible physiological response differences (bleaching before bail-out) compared to the others, corresponding data sets were not included in subsequent analyses. In the hypersaline experiment, a significant difference was found among time points for the Chao1 index (Friedman's rank sum test; $P < 0.05$; Fig. 2), whereas in the hyperthermal experiment, significant differences were identified

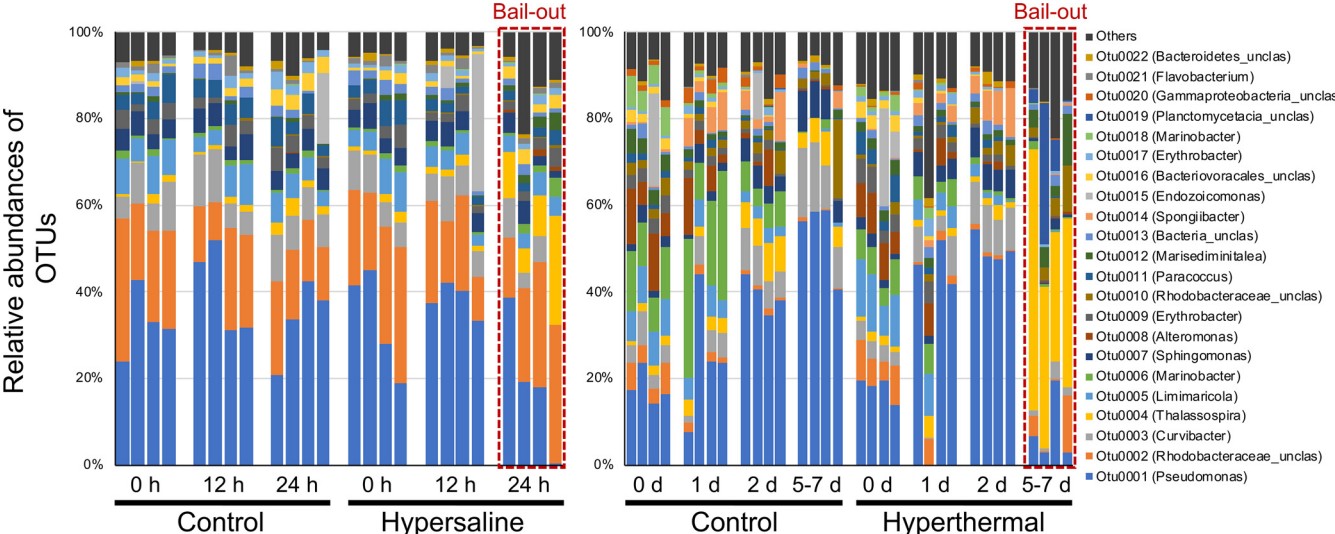

**FIG 3** OTU composition in *P. acuta* in the hypersaline (left) and hyperthermal (right) experiments. Only OTUs with total abundances greater than 0.5% in the whole data set (both experiments) are presented. Samples are arranged in the same order for each condition (*Pocillopora* number 10, number 22, number 23, number 24). Onset of polyp bail-out is highlighted with red boxes.

among time points for the Shannon and inverse Simpson indexes. Neither experiment showed a significant treatment effect. Although for all three indexes higher values were found at the onset of polyp bail-out in both experiments (24 h and days 5 to 7 in the hypersaline and hyperthermal experiments, respectively), none of the pairwise comparisons to the corresponding control groups showed significant differences after FDR adjustment (pairwise Wilcoxon rank sum test). Detailed results of statistical analyses are provided in Table S1 and a corresponding plotting including *Pocillopora* number 15 is provided in Fig. S3.

**Structure of coral-associated bacterial communities and beta diversity. Class level.** When grouping OTUs according to their classes, all non-bail-out *P. acuta* libraries were dominated by *Gammaproteobacteria* (mean $\pm$ standard deviation = 52% $\pm$ 13%) and *Alphaproteobacteria* (33% $\pm$ 12%; Fig. S4; results including *Pocillopora* number 15 are provided in Fig. S5). A decrease in abundance of *Gammaproteobacteria* (23% $\pm$ 11%) and an increase in *Alphaproteobacteria* (57% $\pm$ 15%) were observed in libraries of polyp bail-out in both the hypersaline (24 h) and hyperthermal experiments (days 5 to 7). Two-factor permutational multivariate analysis of variance (PERMANOVA) for class-level bacterial composition identified significant differences among libraries based on both treatment and time in the hyperthermal experiment (two-factor PERMANOVA; $P < 0.05$; Table S2), but neither factor was significant in the hypersaline experiment. A significant interaction between treatment and time was also identified in the hyperthermal experiment. However, when focusing on treatment effect at each time point, none of the pairwise comparisons in either experiment showed significant differences after $P$-value adjustment using the false discovery rate method (FDR-adjusted $P$-value $< 0.05$).

**OTU level.** At the OTU level, OTU0001 (*Pseudomonas* sp., 36% $\pm$ 9%) and OTU0002 (*Rhodobacteraceae*_unclassified, 20% $\pm$ 7%), dominated all non-bail-out *P. acuta* libraries in the hypersaline experiment (Fig. 3; results including *Pocillopora* number 15 are provided in Fig. S6). A discernible decrease in OTU0001 (19% $\pm$ 16%) and an increase in OTU0004 (*Thalassospira* sp., from 2% $\pm$ 1% to 13% $\pm$ 8%) were found at the onset of polyp bail-out (24 h in the treatment group). Significant differences were found among time points in the hypersaline experiment (two-factor PERMANOVA; $P < 0.05$; Table S3), but not between treatments.

In the hyperthermal experiment, non-bail-out *P. acuta* libraries were generally dominated by OTU0001 (34% $\pm$ 17%), but with considerable variation (0 to 59%; Fig. 3). Following OTU0001, various taxa dominated at different time points and conditions, including OTU0002, OTU0003 (*Curvibacter* sp.), OTU0005 (*Limimaricola* sp.), and OTU0006 (*Marinobacter* sp.). As in the hypersaline experiment, onset of polyp bail-out in the hyperthermal experiment (days 5 to

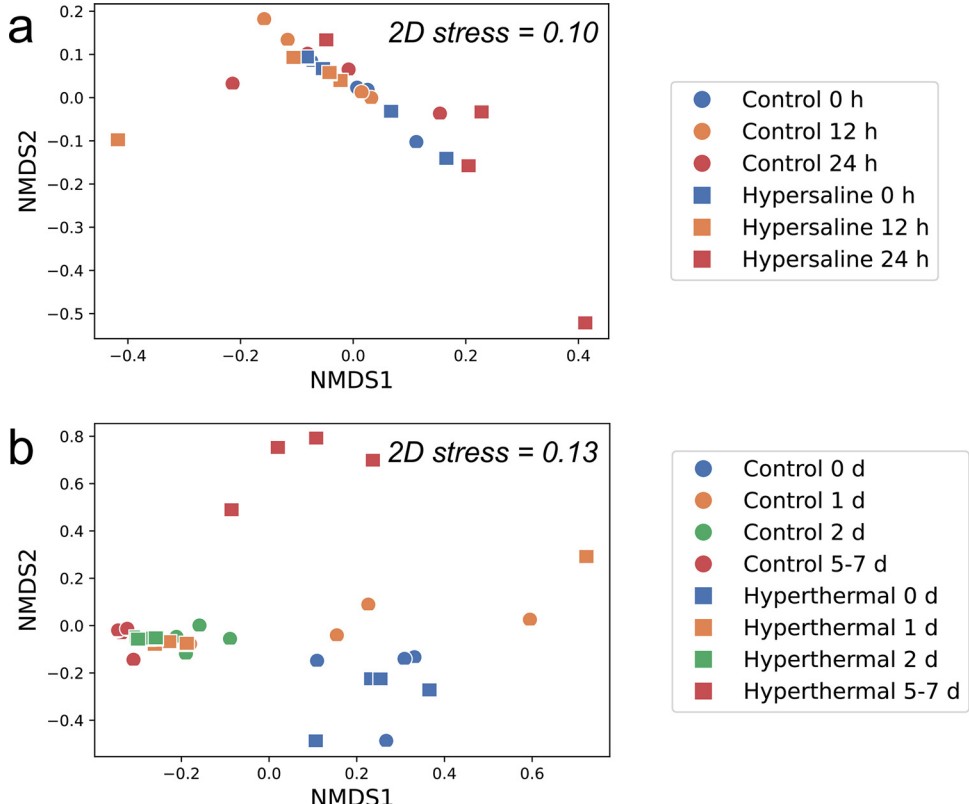

**FIG 4** NMDS visualization of OTU composition in the hypersaline (a) and hyperthermal (b) experiments. Libraries in the treatment and control groups are presented in squares and circles, respectively. Sampling points are highlighted in different colors (N = 4 per condition).

7 in the treatment group) was characterized by a decrease in OTU0001 (8% $\pm$ 8%) and an increase in OTU0004 (from 3% $\pm$ 2% to 42% $\pm$ 13%). Significant differences in OTU composition were found between treatment and control groups and among time points in the hyperthermal experiment, with the interaction between the two factors also being significant (two-factor PERMANOVA; $P < 0.05$; Table S3). When focusing on the treatment effect at individual time points, differences between treatment and control groups were significant throughout the hyperthermal experiment, except on day 1. Consistent with the statistical analysis, nonmetric multidimensional scaling (NMDS) plotting showed clear separation of treatment and control groups at the onset of polyp bail-out in the hyperthermal experiment (days 5 to 7), while the differentiation was less clear in the hypersaline experiment (Fig. 4; results including *Pocillopora* number 15 are provided in Fig. S7). Homogeneity of molecular variance analysis (HOMOVA) identified greater variances at the onset of polyp bail-out in both experiments. However, the differences were not significant after FDR adjustment (Table S4). A complete listing of OTU abundance and taxa is provided in Tables S5 to S7.

**Indicator OTUs at the onset of polyp bail-out.** Using LEfSe, 7 and 40 OTUs were identified as significant indicators for the onset of polyp bail-out in the hypersaline and hyperthermal experiments, respectively (Kruskal-Wallis test; $P < 0.05$; logarithmic LDA score > 2; Fig. S8). Four indicator OTUs were common to both experiments, including OTU0004, OTU0012 (*Marisediminitalea* sp.), OTU0036 (*Rhodobacteraceae*_unclassified), and OTU0080 (*Myxococcales*_unclassified), which all showed clearly higher abundances when polyp bail-out was observed (24 h and days 5 to 7 in the hypersaline and hyperthermal experiments, respectively; Fig. 5).

## DISCUSSION

Polyp bail-out may be a strategy of certain scleractinian corals to facilitate dispersal. Contemporary studies have documented polyp bail-out in response to different stressors

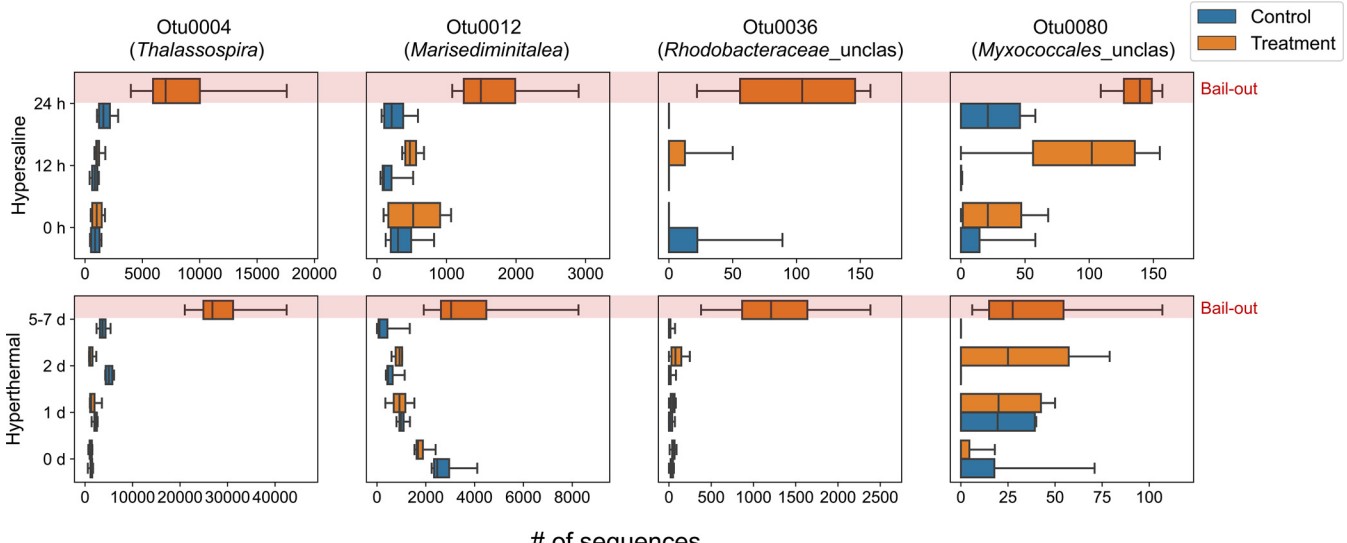

**FIG 5** Abundance changes of OTU0004 (*Thalassospira* sp.), OTU0012 (*Marisediminitalea* sp.), OTU0036 (*Rhodobacteraceae*_unclassified), and OTU0080 (*Myxococcales*_unclassified) in both experiments. Boxes and whiskers indicate the quartiles and full range of the data sets, respectively.

(15, 18, 22, 33–35). In the present study, we conducted two experiments to induce polyp bail-out in *P. acuta*, a scleractinian coral commonly used in polyp bail-out research, and examined bacterial community changes during bail-out induction (21, 31, 33, 36). In both experiments, microbiomes in *P. acuta* corals were dominated by *Proteobacteria* (especially *Gammaproteobacteria* and *Alphaproteobacteria*; Fig. S4), consistent with those identified in field-collected samples (37, 38). At the OTU level, *Pseudomonas* sp. (OTU0001) and an unclassified *Rhodobacteraceae* bacterium (OTU0002) were the dominant bacterial taxa in non-bail-out libraries (Fig. 3). *Pseudomonadaceae* and *Rhodobacteraceae* predominate in *P. acuta* in Singapore and the Great Barrier Reef (38–40). Several marine isolates of *Pseudomonas* possess antimicrobial, antifungal, or biodegrading activity (41, 42). Although evidence is yet not available for scleractinian corals, *Pseudomonas* bacteria isolated from the soft corals *Sarcophyton glaucum* and *Sinularia polydactyla* inhibit growth of some other bacteria and fungi and are thought to protect coral hosts against pathogens (43, 44). Our results add further evidence for the association of *Pseudomonas* bacteria with *P. acuta*, warranting further investigation of its ecological functions in coral holobionts. On the other hand, *Endozoicomonas*, another bacterial clade reportedly predominant in *P. acuta* (37, 45) and other *Pocillopora* corals (46, 47), presented only sporadically in our coral samples (Fig. 3). As microbiomes in *P. acuta* vary even within short geographic distances (38, 40), inconsistency of our results with other studies is not surprising and can likely be attributed to site-specific variation in *P. acuta* microbiomes.

In this study, we observed CAB changes as a temporal response in both our experiments (especially the hyperthermal experiment; Fig. 3). Given that culture conditions in our indoor aquaria (closed system; lower light intensity; artificial seawater) differed from those in which corals were acclimated before experiments (open system; natural sunlight; sand-filtered natural seawater), CAB changes during cultivation are not unexpected. In contrast to the time effect, the hypersaline treatment exerted no significant effect on microbiomes of *P. acuta*. Addition of hypersaline seawater in the treatment aquarium could have introduced exotic bacteria and represent another possible factor in the treatment effect (no water was added to the control aquarium). Nevertheless, our findings suggest a negligible effect from addition of water during 24 h. These results are also consistent with that reported in *F. granulosa*, in which microbiomes remained stable after short-term (4 h) exposure to hypersaline stress (14). Unfortunately, as studies about effects of hypersaline stresses on coral microbiomes are limited, a solid conclusion cannot be drawn without further investigation.

On the other hand, a significant treatment effect was identified in our hyperthermal experiment (Fig. 3). Thermal stresses can strongly influence coral physiology and may

alter microbiomes in stony corals. Decreased *Endozoicomonas* and increased *Rhodobacteraceae* or *Vibrio* have frequently been found in thermally stressed corals (48–54). However, these changes were not found in our corals, reflecting a species- or location-specific response in our *P. acuta*. Another potential effect from the hyperthermal treatment is stronger evaporation at elevated temperatures, which caused higher salinity fluctuations in the treatment aquarium. Meanwhile, to compensate for stronger evaporation, higher volumes of fresh water were added to the treatment aquarium, representing another possible factor in differences between the treatment and control groups in our hyperthermal experiment. It is also worth noting that bacterial communities showed a significant difference between treatment and control groups even at the beginning of our hyperthermal experiment (day 0 in Fig. 3). The significant effect of hyperthermal treatment in this study therefore may also be attributed to a cage effect in our experimental setup. As bacterial composition in seawater was not examined in this study (due to the small size and limited number of aquaria), how water addition and cage effects influenced coral microbiomes remains unclear. Nevertheless, given that bacterial communities in treatment and control groups were not significantly different at the class level (per time point; Fig. S4), observed variations at the OTU level probably reflect a shift between functionally redundant bacterial groups without significant detriment to coral physiology, as suggested by Lu et al. (55) and Hernandez-Agreda et al. (56).

In the LEfSe analysis, four indicator OTUs of polyp bail-out, affiliated with *Thalassospira* (OTU0004), *Marisediminitalea* (OTU0012), *Rhodobacteraceae* (OTU0036), and *Myxococcales* (OTU0080), occurred in all our experiments (Fig. 5 and Fig. S8). A BLAST search of OTU0004 in NCBI (rRNA/ITS databases) yielded two *Thalassospira* species, *T. xiamenensis* (GenBank accession number: NR_042780.1; 99.31% gene identity) and *T. permensis* (NR_116841.1; 99.31% gene identity). *Thalassospira* bacteria were found in association with healthy corals such as *Cladocora caespitosa*, *Millepora alcicornis*, and *Sinularia* sp. (57–60) and genetic studies have suggested their possible roles in phosphorus and iron cycles (57, 61). On the other hand, OTU0012, OTU0036, and OTU0080 best matched *Marisediminitalea aggregata* (NR_116838.1; 98.97% gene identity), *Allosediminivita pacifica* (NR_126266.1; 98.63% gene identity), and *Sandaracinus amylolyticus* (NR_118001.1; 93.15% gene identity), respectively. Common increases of these bacteria in both our experiments imply their involvement in development of polyp bail-out. Although, to our knowledge, no previous study has reported increases of these bacterial taxa in thermally or osmotically stressed corals, species- and site-specific differences in coral microbiomes may also contribute. The possibility of opportunistic growth of these bacteria thus cannot be fully excluded. It should also be mentioned that due to the design of this study (limited size of samples, only one aquarium for each treatment group, etc.), the present results should be interpreted with caution. These findings, however, provide a basis to test the hypothetical involvement of microbes in polyp bail-out. Future studies of the capacity of specific bacteria to induce polyp bail-out as pure isolates or a community may yield a clearer picture of microbial role in polyp bail-out.

Notably, in *P. damicornis* (*Pocillopora* number 15) we observed bleaching on day 3 in the hyperthermal experiment, prior to the occurrence of polyp bail-out (Fig. S1). However, bleaching was not observed in our *P. acuta* corals (Fig. 1), suggesting different resistance to bleaching in the two coral species. In an earlier study, cooccurrence of bleaching and polyp bail-out was observed in *P. damicornis* under hyperthermal stress (62). Interestingly, bailed-out polyps in the same study showed less clear morphological differentiation, which was also observed in some of our bleached, bailed-out polyps from *P. damicornis*. This undifferentiated polyp morphology resembles that of degenerated polyps reported in Chuang et al. (21), implying possible physiological damage in bailed-out polyps by hyperthermal stress. Given that polyp recovery after bail-out is likely an energetically demanding process, thermal bleaching must significantly reduce recovery of bailed-out polyps, compromising their resettlement capacity. The relative developmental speeds of bail-out and bleaching, therefore, may have strong implications for dispersal potential of corals against the specter of future climate change.

## MATERIALS AND METHODS

**Coral sampling.** In summer 2021, we obtained several *Pocillopora* corals from the Onna fisheries association in Okinawa, Japan, which collected corals from the ocean near Onna Village on the western side of Okinawa. Collected corals were transferred to an outdoor, open-system aquarium (3000 L) at the OIST Marine Science Station at Seragaki, Okinawa. Sand-filtered seawater pumped from the ocean was supplied to the outdoor aquarium. All coral colonies were acclimated for over 6 months prior to experiments.

**Polyp bail-out induction and coral tissue sampling.** Five healthy colonies were employed in experiments in the present study. Induction of polyp bail-out was accomplished using hypersaline and hyperthermal methods, modified from Chuang and Mitarai (31) and Gösser et al. (33), respectively. One day prior to each experiment, small fragments (~1 cm branch tips; six fragments/colony for the hypersaline experiment and eight fragments/colony for the hyperthermal experiment) were separated from selected colonies using a clean bone cutter and placed in two 5-L indoor aquaria (one denoted as the treatment group and the other denoted as the control group). Each aquarium was equipped with a 3.4 W slim filter pump (GEX, Japan) for water circulation. Seawater was prepared at 35‰ by dissolving artificial sea salt (Kaisuimaren, Japan) in reverse osmosis (RO) water. Light was provided at 50 $\mu$mol photons/m$^2$/s using a Mitras Lightbar 2 (GHL, Germany) with a 12-h light-dark cycle (light: 06:00 to 18:00) and seawater temperature was set at 25℃ in both aquaria using two 110W aquarium heaters (Kotobuki Kogei, Japan). Seawater salinity and temperature were monitored with a ProfiLux 4 aquarium controller (GHL, Germany).

**Hypersaline experiment.** Hypersaline artificial seawater was prepared in a separate aquarium at 49‰ and supplied to one of the 5-L aquaria (treatment group) at 4 mL/min, while no water was added to the other aquarium (control group). An overflow design maintained a fixed water volume in both aquaria during the hypersaline experiment. The experiment was started at 10:00 a.m. From both treatment and control groups, one fragment from each colony was collected using a clean forceps at 0 h, 12 h, and 24 h (onset of polyp bail-out, defined as distinguishable colony dissociation and polyp detachment; Fig. 1), making a total of 5 fragments per condition per time point. Collected coral fragments were immediately placed in sterile 1.5 mL Eppendorf tubes and stored at −20℃. For each sample of polyp bail-out, detached polyps (collected using a micropipette and a sterile pipette tip with <100 $\mu$L seawater) and the remaining skeleton (with undetached polyps, if present; collected using a clean forceps) were collected as one sample to keep the sampling consistent during the experiment.

**Hyperthermal experiment.** After 1 day of acclimation at 25℃, one aquarium was warmed by 3℃ every day (conducted at 10:00 am; treatment group), while the other aquarium was maintained at 25℃ (control group). It took several minutes to reach the set temperature. As polyp bail-out was reported at 34℃ in a previous study (33), the treatment group was warmed to 34℃ and was maintained at that temperature thereafter. One fragment from each colony was collected using a clean forceps in both treatment (before temperature changes) and control groups at day 0, day 1, day 2, and when polyp bail-out was observed (days 5 to 7; Fig. 1), equaling 5 fragments per condition per time point. To compensate for water evaporation during the experiment (salinity fluctuation <1% in the control group and <3% in the treatment group at elevated temperatures), fresh RO water was added to both aquaria every day (after tissue sampling) to a fixed water level. As in the hypersaline experiment, both detached polyps and skeletons were collected for samples of polyp bail-out. Collected coral samples were immediately stored at −20℃ until further processing.

**DNA extraction, species identification, and 16S-rRNA gene sequencing.** In total, 70 coral tissue samples were collected, including 30 from the hypersaline experiment (5 colonies × 2 conditions × 3 time points) and 40 from the hyperthermal experiment (5 colonies × 2 conditions × 4 time points). Total DNA from collected tissue samples was extracted using a DNeasy blood and tissue kit (Qiagen, Japan) following the manufacturer's instructions. To focus on the microbiome in coral tissues and the surface mucus layer, skeletons were not ground and were removed after attached tissues were all lysed in tissue lysis buffer. Two tissue-negative controls (containing only tissue lysis buffer) were included in the DNA extraction step. DNA concentrations were checked using a Qubit 4 Fluorometer (Thermo Fisher Scientific, Japan). For each coral colony, the DNA sample at 0 h in the control group of the hypersaline experiment was used for genotyping coral species. The mitochondrial region was amplified using the primer pair FATP6.1/RORF and PCR conditions reported in Flot et al. (63). PCR was conducted using AmpliTaq Gold 360 Master Mix (Thermo Fisher Scientific, Japan) following the product manual. PCR products were submitted to Macrogen, Japan for Sanger sequencing from both ends. Sequencing results were aligned with reference sequences (accession numbers: KY587458-KY587472; JX985584-JX985620; EU374225-EU374261) reported in previous studies (63–65). Coral species were identified by phylogenetic analyses based on both Neighbor-Joining and Maximum Likelihood trees, constructed using substitution models reported in Schmidt-Roach et al. (66) and Poquita-Du et al. (65), respectively. All phylogenetic analyses were conducted using MEGA X, with 1,000 bootstrap pseudoreplications (67). To examine microbiome composition in the corals, DNA samples (including the negative controls) were submitted to paired-end (2 × 300 bp) Illumina MiSeq (v3) sequencing. The primer pair specific for the V5-V6 region of the 16S-rRNA gene (784F: 5′-TCGTCGGCAGCGTCAGATGTGTATAAGAGACAGAGGATTAGATACCCTGGTA-3′′; 1061R: 5′-GTCTCGTGGG CTCGGAGATGTGTATAAGAGACAGCRRCACGAGCTGACGAC-3′) was used to construct 16S-rRNA gene libraries, given their utility in detecting bacterial communities associated with corals (14, 68, 69). Preparation of 16S-rRNA gene libraries and subsequent sequencing were conducted by the Sequencing Section at the Okinawa Institute of Science and Technology (OIST), Japan.

**16S-rRNA sequencing data processing.** We used Mothur version 1.44.1 (70) to analyze sequencing data. Raw data were first assembled into contigs using the "make.contigs" command in Mothur with default parameter settings, which assembles paired sequencing reads into contigs according to the corresponding quality scores in fastq files. Given a median length of 308 bp in our assembled contigs, assembled contigs longer than 350 bp or shorter than 250 bp were filtered out using the "screen.seqs" command with the parameters: minlength = 250 and maxlength = 350, assuming they represent incorrect/unsuccessful assembly. Criteria

for ambiguous base and homopolymer filtering were set as suggested by the Mothur development team (maxambig = 0, and maxhomop = 8). Duplicated contigs were removed using the "unique.seqs" command and remaining contigs were aligned against the SILVA database v138_1 (71). Contigs that differed by ≤3 bp were clustered using the "pre.cluster" command, assuming about 1% error in the sequencing data. Chimeric contigs were removed using the "chimera.vsearch" command. Taxonomy of remaining contigs was determined using an RDP database (version 18) at a cutoff value of 80 and nonbacterial contigs were removed. Decontamination was then conducted using microDecon (72), software that assumes fixed ratios among contaminant contigs in blank (tissue-negative controls in this study) and true samples and removes them from true samples using a proportion-based approach. Parameters in microDecon were set as the default, except for "thresh," which was set to 1 to accommodate genotype-specific contigs. After decontamination, contigs of ≥97% similarity were grouped into OTUs and consensus taxa were assigned. For further analyses, coral tissue libraries were rarefied by randomly subsampling 70,259 contigs (the minimum among all libraries; Good's coverage >99%) in each library. Bacterial alpha diversity in each library was calculated for the Chao1, inverse Simpson, and Shannon indexes following instructions in Mothur. Nonmetric multidimensional scaling (NMDS) plotting was conducted independently for the hypersaline and hyperthermal experiments to visualize beta diversity among conditions (treatments and time points) in each experiment.

**Statistical analyses.** Statistical analyses for alpha and beta diversity were conducted in the R environment (v4.2.3) or Mothur (v1.44.1). Given that samples collected at each time point in this study represented only pseudoreplicates (only one aquarium was used for each treatment in our experiments) and were repeatedly sampled from the same colonies, nonparametric statistical analyses were applied. Friedman's rank sum test was conducted to examine statistical differences in alpha diversity among conditions (treatments or time points) for each experiment. Although Friedman's test does not test for interactions between factors, *post hoc* analysis for alpha diversity was conducted using the pairwise Wilcoxon rank sum test for all possible combinations of conditions (treatments × time points) to allow examination of differences between treatment and control groups at specific time points. Statistic differences in bacterial composition among conditions in each experiment were tested using two-factor permutational multivariate analysis of variance (two-factor PERMANOVA; 999 permutations) with the adonis2() function in the R package, vegan, with subsequent multiple pairwise comparisons conducted using the pairwise.adonis2() function. Differences in data variance between conditions (treatments × time points) in each experiment were examined by homogeneity of molecular variance analysis (HOMOVA; 1,000 permutations), conducted in Mothur. To identify indicator OTUs responsible for polyp bail-out, an LDA Effect Size (LEfSe) analysis was carried out for coral tissue libraries at the time of polyp bail-out in both experiments (24 h in the hypersaline experiment; 5 to 7 day in the hyperthermal experiment) using default settings (73). Differences were considered significant at $P < 0.05$ (FDR-adjusted *P*-values for multiple comparisons) for all statistical analyses.

**Data availability.** Raw MiSeq sequencing data have been uploaded to the NCBI Sequence Read Archive (SRA) under BioProject PRJNA906327.

## SUPPLEMENTAL MATERIAL

Supplemental material is available online only.
**SUPPLEMENTAL FILE 1**, TIF file, 4.1 MB.
**SUPPLEMENTAL FILE 2**, TIF file, 1.4 MB.
**SUPPLEMENTAL FILE 3**, TIF file, 0.6 MB.
**SUPPLEMENTAL FILE 4**, TIF file, 2.1 MB.
**SUPPLEMENTAL FILE 5**, TIF file, 2.4 MB.
**SUPPLEMENTAL FILE 6**, TIF file, 2.6 MB.
**SUPPLEMENTAL FILE 7**, TIF file, 0.9 MB.
**SUPPLEMENTAL FILE 8**, TIF file, 1.9 MB.
**SUPPLEMENTAL FILE 9**, XLSX file, 0.5 MB.

## ACKNOWLEDGMENTS

This work was supported by JSPS KAKENHI (Grant number: JP20K19960) and JST FOREST Program (Grant number: JPMJFR2070). We gratefully acknowledge generous support to the Marine Biophysics Unit from Okinawa Institute of Science and Technology Graduate University (OIST). We thank the OIST DNA Sequencing Section for 16S-rRNA gene library construction and sequencing and the OIST Scientific Computing and Data Analysis Section for providing the high-performance computing service for data analysis. We thank Steven D. Aird for editing and commenting on the manuscript. We thank Chih-Ying Lu for technical support on data analysis.

We have no conflicts of interest to declare.

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
