## [Reviewer comments · Microbiology Spectrum]

Microbiology Spectrum

Bacterial community shifts during polyp bail-out induction in *Pocillopora* corals

Po-Shun Chuang, Yosuke Yamada, Po-Yu Liu, Sen-Lin Tang, and Satoshi Mitarai

Corresponding Author(s): Po-Shun Chuang, Biodiversity Research Center Academia Sinica

Review Timeline:

Submission Date:	January 16, 2023
Editorial Decision:	March 9, 2023
Revision Received:	April 19, 2023
Editorial Decision:	May 14, 2023
Revision Received:	June 5, 2023
Accepted:	June 8, 2023

Editor: Nikki Traylor-Knowles

Reviewer(s): Disclosure of reviewer identity is with reference to reviewer comments included in decision letter(s). The following individuals involved in review of your submission have agreed to reveal their identity: Shumpei Maruyama (Reviewer #2)

Transaction Report:

DOI: <https://doi.org/10.1128/spectrum.00257-23>

March 9, 2023

Dr. Po-Shun Chuang
Biodiversity Research Center Academia Sinica
Taipei
Taiwan

Re: Spectrum00257-23 (Bacterial community shifts during polyp bail-out induction in *Pocillopora* corals)

Dear Dr. Po-Shun Chuang:

The reviewers found this work interesting, however there are significant revisions that are needed for publication including more flushed out methods and a suggestion that the results are overstated.

Link Not Available

Sincerely,

Nikki Traylor-Knowles

Journals Department
Reviewer comments:

Reviewer #1 (Public repository details (Required)):

Sequences were deposited in ncbi

Reviewer #1 (Comments for the Author):

Spectrum Review

This study was an examination into the microbial changes surrounding polyp bail out. Polyp bail out is an increasingly interesting area of study, as it is an unusual mechanism for corals to withstand stressful conditions. The authors posit that microbiome may cause polyp bail out. They showed some of the microbial changes associated with polyp bail out after it is induced by a thermal

or hypersaline challenge. The implications of these changes require further study, but are interesting and potentially exciting. There are details missing from the experiment, methods and statistics, which I have outlined below. Additionally, the results should include more information about when the polyp bail out occurred, and the images of the corals throughout the time series. The discussion requires areas of clarification and I have indicated locations where the implications of the data are overstated.

Major and minor comments by section and line:

Line 120: define TNF and TNFR

Results

It is not clear in the figures which samples were from different species, and how the different species may have influenced the data and interpretation of the results.

Line 167: On the plot there were multiple time points indicated with significant differences between treatment and control in terms of diversity, which this statement suggested.

Line 171: "Polyp bail out was associated with the increase in alpha diversity" does not make sense with the earlier statement that there was no difference in alpha diversity except on day 2

- It would be helpful to have a comparison of before vs after polyp bail out to support this statement

Line 183 & in Fig 2- indicate when polyp bail out occurred

Discussion:

Line 262: results show a decrease in *Pseudomonas* in the treatment and an increase in the controls - it is not clear how these results indicate time dependent establishment and resilience.

Line 264: "elevated seawater temperatures seem to facilitate microbiome re-establishment in *Pocillopora* corals, as observed in our hyperthermal experiment." What is the evidence for re-establishment?

Line 278: I disagree with this statement. Because the temperature and salinity were still being changed during the timing of polyp bail out, it is not clear if it is environmentally mediated microbial changes or host-induced microbial changes. Because no host-level data is shown, it is also difficult to understand the other physiological changes associated with increased temperature.

Line 292: Opportunistic growth cannot be ruled out. For example, there are no water samples or environmental samples included, and there are site-specific differences in microbiomes, which may be why the taxa observed in this study were not observed in other studies. Right now these observations are correlative and not causative and should not be stated as causative.

The discussion should include what microbial taxa are typically associated with *Pocillopora* in other studies and have been known to change with bleaching and salinity.

Methods

How many tanks were used for each experiment?

How many fragments were removed for sampling during each time point?

What types of heaters were used to raise the temperature? How temperature kept constant with in the tank?

How were coral samples taken for microbial analysis?

What were the sampling and procedural controls for the 16S analysis?

How many libraries were constructed? If more than one, how were the different libraries combined?

Line 339: Not clear if the coral was sampled or the polyp that had bailed was sampled for 16S

Line 375: Why was the V5-V6 region used? The V4 region is usually used in coral microbiome studies.

Data processing:

How was error and quality parameters decided?

When were paired end reads merged? Is that is what is meant as "assembling contigs" - which is typically not language used in 16S studies. Why was merging conducted before quality control of sequences?

Statistics

It seems as though the question the authors are asking is how did the microbiome of corals change before and after polyp bail out - this does not seem to be what was tested statistically, or it is not clear what was tested statistically to demonstrate that effect. Models should include Treatment, Time, before/after bail out, and an effect for the coral sampled (or colony sampled).

Line 403: comparison across libraries is not clear statement is not clear. What treatments/days were compared for the PERMANOVA? How were multiple comparisons of the data taken into account? What programs were used to perform the PERMANOVA?

Line 405: What does the metastats command do in mothur? Specify what this analysis is, what the assumptions are and how it is run.

Figures/Tables

Table 1: Does >1% refer to the entire dataset? Are these in the top 1% of either experiment or across both experiments (i.e., when the data are combined).

Figures should include when polyp bail out occurred to understand the comparisons made.

Why are both Figures 2 and Figure 2 shown? The taxa information could be included in figure 3 and grouped by class and that would provide the same information.

Figure 4 and 5: Indicate taxonomic information (e.g. Genus) with OTUs in figures 4 and 5

Figure 6: indicate the number of samples, and how many corals exhibited polyp bail out at each time point.

Reviewer #2 (Public repository details (Required)):

They have 16s microbiome datasets which have been submitted to NCBI already.

Reviewer #2 (Comments for the Author):

Please see attached document.

Staff Comments:

Preparing Revision Guidelines

Please return the manuscript within 60 days; if you cannot complete the modification within this time period, please contact me. If you do not wish to modify the manuscript and prefer to submit it to another journal, please notify me of your decision immediately so that the manuscript may be formally withdrawn from consideration by Microbiology Spectrum.

Spectrum Review

This study was an examination into the microbial changes surrounding polyp bail out. Polyp bail out is an increasingly interesting area of study, as it is an unusual mechanism for corals to withstand stressful conditions. The authors posit that microbiome may cause polyp bail out. They showed some of the microbial changes associated with polyp bail out after it is induced by a thermal or hypersaline challenge. The implications of these changes require further study, but are interesting and potentially exciting. There are details missing from the experiment, methods and statistics, which I have outlined below. Additionally, the results should include more information about when the polyp bail out occurred, and the images of the corals throughout the time series. The discussion requires areas of clarification and I have indicated locations where the implications of the data are overstated.

Major and minor comments by section and line:

Line 120: define TNF and TNFR

Results

It is not clear in the figures which samples were from different species, and how the different species may have influenced the data and interpretation of the results.

Line 167: On the plot there were multiple time points indicated with significant differences between treatment and control in terms of diversity, which this statement suggested.

Line 171: “Polyp bail out was associated with the increase in alpha diversity” does not make sense with the earlier statement that there was no difference in alpha diversity except on day 2

- It would be helpful to have a comparison of before vs after polyp bail out to support this statement

Line 183 & in Fig 2- indicate when polyp bail out occurred

Discussion:

Line 262: results show a decrease in *Pseudomonas* in the treatment and an increase in the controls – it is not clear how these results indicate time dependent establishment and resilience.

Line 264: “elevated seawater temperatures seem to facilitate microbiome re-establishment in *Pocillopora* corals, as observed in our hyperthermal experiment.” What is the evidence for re-establishment?

Line 278: I disagree with this statement. Because the temperature and salinity were still being changed during the timing of polyp bail out, it is not clear if it is environmentally mediated microbial changes or host-induced microbial changes. Because no host-level data is shown, it is

also difficult to understand the other physiological changes associated with increased temperature.

Line 292: Opportunistic growth cannot be out ruled. For example, there are no water samples or environmental samples included, and there are site-specific differences in microbiomes, which may be why the taxa observed in this study were not observed in other studies. Right now these observations are correlative and not causative and should not be stated as causative.

The discussion should include what microbial taxa are typically associated with *Pocillipora* in other studies and have been known to change with bleaching and salinity.

Methods

How many tanks were used for each experiment?

How many fragments were removed for sampling during each time point?

What types of heaters were used to raise the temperature and what type of thermometer was used? How temperature kept constant with in the tank?

How were coral samples taken for microbial analysis?

What were the sampling and procedural controls for the 16S analysis?

How many libraries were constructed? If more than one, how were the different libraries combined?

Line 339: Not clear if the coral was sampled or the polyp that had bailed was sampled for 16S

Line 375: Why was the V5-V6 region used? The V4 region is usually used in coral microbiome studies.

Data processing:

How was error and quality parameters decided?

When were paired end reads merged? Is that is what is meant as “assembling contigs” – which is typically not language used in 16S studies. Why was merging conducted before quality control of sequences?

Statistics

It seems as though the question the authors are asking is how did the microbiome of corals change before and after polyp bail out – this does not seem to be what was tested statistically, or it is not clear what was tested statistically to demonstrate that effect. Models should include Treatment, Time, before/after bail out, and an effect for the coral sampled (or colony sampled).

Line 403: comparison across libraries is not clear statement is not clear. What treatments/days were compared for the PERMANOVA? How were multiple comparisons of the data taken into account? What programs were used to perform the PERMANOVA?

Line 405: What does the metastats command do in mothur? Specify what this analysis is, what the assumptions are and how it is run.

Figures/Tables

Table 1: Does >1% refer to the entire dataset? Are these in the top 1% of either experiment or across both experiments (i.e., when the data are combined).

Figures should include when polyp bail out occurred to understand the comparisons made.

Why are both Figures 2 and Figure 2 shown? The taxa information could be included in figure 3 and grouped by class and that would provide the same information.

Figure 4 and 5: Indicate taxonomic information (e.g. Genus) with OTUs in figures 4 and 5

Figure 6: indicate the number of sample, and how many corals exhibited polyp bail out at each time point.

The authors of this manuscript sought to characterize members of the microbiome that may be responsible for polyp bailout. While they did not empirically test candidate microbes that were identified in their study and their role in polyp bailout, it is a novel and important study that can begin to test this hypothesis. Overall I believe the study is effective in its approach and they successfully identified microbes that are associated with polyp bailout in *Pocillopora* spp. However, I have several major and minor concerns with the methodology that should be addressed.

Overall, there needs to be more detail in the methods portion of this manuscript. I have outlined my concerns below, in no particular order of importance:

1. Line 326 - is 50 umol of light used in the indoor aquaria comparable to the outdoor light settings? It does not appear likely. Please inform the reader how light could potentially alter the microbiome (if at all).
2. Of the corals used in this study, 1 out of 5 of the coral was *P. damicornis*. It is unclear if the microbiome of *P. damicornis* is comparable to that of *P. acuta* as they are entirely different species. I believe that the microbial dataset for *P. damicornis* should be removed as it could be a confounding variable, or that it should be justified why it was kept in the study and the figures should clearly denote which sample came from *P. damicornis* as it is not obvious which one it may be, especially as some coral microbiomes appear to behave differently to others. For example, the microbial community from the fifth bar in Figure 4 has a dramatically different abundance of OTU0015 compared to the others - is that the *P. damicornis* sample?
3. Were the corals kept in aquaria with no flow or filter for 7 days during your heat-stress experiment? Please clarify the aquarium setup.
4. Line 333 - Water was added to the hypersaline treatment to slowly increase the salinity, however the control treatment had no additional seawater added. This is a confounding variable and should be justified.
5. Line 333 - In addition, please add information on what the source of your artificial seawater is (what salt manufacturer) and whether it was sterilized with a filter and/or autoclaving.
6. Line 350 - What is the source of the freshwater in the heat-stress experiment? Tap water? DI water? Was this sterilized as well? Was the salinity fluctuation significant?
7. Line 359 - Please clarify what was used in the negative control for the PCR. Nuclease-free water from the kit? Or seawater from your seawater source?
8. An important concern: It appears that only two 5L aquaria were used for each of the hypersaline and heat stress experiments. This could cause significant cage effects on the microbial community (especially from coprophagy) and should be clearly justified and clarified in the discussion section. If this was not the case, and each coral fragment was split into their own respective aquaria (for a total of 30 aquariums for hypersaline and 40 aquariums for heat stress), please clarify.
9. How is data from days 5-7 of the heat stress experiment compiled? Are the coral only sampled after polyp bailout? Or were all genets sampled at both day 5 and day 7? On all figures it appears that days 5-7 were somehow integrated together and it is unclear how that was done.

10. All OTUs present in the negative controls should be removed from your analysis. Especially OTU0001 as the abundance of reads in the negative controls were in the same magnitude as the coral samples. Fortunately the other OTUs of interest (0004, 0012, and 0039) do not appear to be present in the negative control so it should not detract from the overall conclusion of the manuscript. If the authors choose to keep OTU0001 in their analysis, it should be justified, and included in Figure 5 as well. Please see <https://doi.org/10.3389/fmicb.2022.1007877> for best practices on treating PCR contamination data.
11. How did you ensure that equal sampling of the coral skeleton and polyps occurred in the polyp bailout and the non-polyp bailed out samples occur? Was the entire 1cm fragment processed? This portion needs more details, as the differences in microbiomes could potentially be explained by sampling differences post-bailout (for example, if your OTUs of interest are over-represented in the polyp compared to the skeleton).

I also have several minor concerns:

1. Lines 82-95 seem too speculative and do not contribute to the overall introduction of polyp bailout.
2. Figure 6 should be the first figure in the manuscript and visually denote when polyp bailout occurs. Figure 1 can be brought up in Line 142.
3. Figures 1-4, visually labeling the individual coral genets and being consistent with the placement of the genet would be helpful. Also, visually annotating when polyp bailout occurs for each genet would be immensely helpful. And please change “treatment” to “hypersaline” and “hyperthermal” for clarity (like it is done in Figure 1)
4. Line 185 - Two p-values are listed but both are labeled as hypersaline.
5. Figure 3 - please remove the “_” from control and treatment labels.
6. Figure 3 - this dataset illustrates the Anna Karenina principle well and should be included in the discussion that could also potentially explain polyp bailout (see: <https://doi.org/10.1038/nmicrobiol.2017.121>)
7. Figure 5 legend requires information on statistical tests and what the asterisk denotes.

Spectrum Review

This study was an examination into the microbial changes surrounding polyp bail out. Polyp bail out is an increasingly interesting area of study, as it is an unusual mechanism for corals to withstand stressful conditions. The authors posit that microbiome may cause polyp bail out. They showed some of the microbial changes associated with polyp bail out after it is induced by a thermal or hypersaline challenge. The implications of these changes require further study, but are interesting and potentially exciting. There are details missing from the experiment, methods and statistics, which I have outlined below. Additionally, the results should include more information about when the polyp bail out occurred, and the images of the corals throughout the time series. The discussion requires areas of clarification and I have indicated locations where the implications of the data are overstated.

Response: Thank you for your comments. We have now incorporated photos and sample numbers in Fig. 1 and highlighted the timing of polyp bail-out in all figures (except for the NMDS plots). Revisions to the contents of the manuscript are listed point-by-point below.

Revision: Fig.1

Major and minor comments by section and line:

Line 120: define TNF and TNFR

Response: Full names of the two genes have now been added to the sentence.

Revision: “Asynchronous expression of tumor necrosis factor (TNF) and tumor necrosis factor receptor (TNFR) genes in *Pocillopora* corals during bail-out induction led to a hypothesis of a microbial trigger for initiation of polyp bail-out”

Results

It is not clear in the figures which samples were from different species, and how the different species may have influenced the data and interpretation of the results.

Response: Thank you for calling our attention to this issue. Because one of our samples was identified as *P. damicornis* (*Pocillopora* #15) and showed a clearly different response in our hyperthermal experiment (bleaching before bail-out), we removed it from the analyses. New Figures 2-6 now only show results for *P. acuta* (N=4), while analytical results including *Pocillopora* #15 are provided in Supplementary Figures (S3-6). In general, excluding *P. damicornis* doesn't change the conclusion in this study. An appropriate description has been added to the **Results** section.

Revision: “Given that *Pocillopora* #15 was identified as *P. damicornis* and showed discernible physiological response differences (bleaching before bail-out) compared to the others, the corresponding datasets were not included in subsequent analyses (analyses including *Pocillopora* #15 are provided in Fig. S3-6).”

Line 167: On the plot there were multiple time points indicated with significant differences between treatment and control in terms of diversity, which this statement suggested.

Response: In the original figure we denoted statistical significances with asterisks (*) and items of marginal significance were indicated with *p*-values. This presentation style caused some confusion. We now only label statistical significance in Fig. 2 (for $p < 0.05$). In addition, based on Reviewer 2's comments, we conducted decontamination (using microDecon) and removed *P. damicornis* from the analyses. In the new dataset (decontaminated; *P. damicornis* excluded), only the Chao1 index at days 5-7 in the hyperthermal experiment showed a significant difference between the treatment and control groups (labelled with an asterisk). Descriptions have been modified in the revised manuscript.

Revision: “In both experiments, significant differences were found among time points for the Chao1, Shannon, and inverse Simpson indexes (two-way ANOVA; $p < 0.05$), but not between treatment and control groups. A significant interaction between time and treatment was identified for the Chao1 and Shannon indexes in the hyperthermal experiment (Supplementary File 1). While focusing on treatment effects at individual time points, only the Chao1 index at the time point of polyp bail-out in the hyperthermal experiment (days 5-7) showed a significant difference between the treatment and control groups (Fig. 2).”

Line 171: “Polyp bail out was associated with the increase in alpha diversity” does not make sense with the earlier statement that there was no difference in alpha diversity except on day 2 - It would be helpful to have a comparison of before vs after polyp bail out to support this statement

Response: Using the updated datasets (decontaminated; *P. damicornis* excluded), we identified significant time effects regarding all the three alpha diversity indexes in both experiments. A comparison of “before vs. after” polyp bail-out therefore includes both a time effect and a bail-out effect. Since in this study we sought to examine microbiome dynamics during induction of polyp bail-out, we believe that pairwise comparisons at each time point better follow the logic of this study than pooling time points according to the onset of polyp bail-out. Using a Tukey's honestly significant difference (HSD) test, we only identified a significant difference ($p < 0.05$) in Chao1 index at the onset of polyp bail-out in the hyperthermal experiment. While, for reference only (data not shown), we did run a Student's t-test by pooling before-bail-out libraries per condition and compared differences between treatment and control groups for before and after polyp bail-out (control_before-bail-out vs. treatment_before-bail-out, control_after-bail-out vs. treatment_after-bail-out). No differences between treatment and control groups were found before bail-out regarding the three indexes. For the after bail-out comparison, only in the hyperthermal experiment did the Chao1 index show a significant difference between treatment and control groups, a result consistent with those of the two-way ANOVA and Tukey's HSD test. We have included a **Statistical analyses** subsection in the **Materials and Methods** section to describe statistical analyses conducted in this study in detail.

Revision: “Statistical analyses for alpha and beta diversity were conducted in the R environment (v3.6.1). Two-way analysis of variance (two-way ANOVA) was conducted using the function `avov()` to examine statistical differences in alpha diversity among conditions (treatments and time points) for each experiment, followed by a Tukey's honestly significant difference (HSD) test for post-hoc analysis. Statistic differences in bacterial composition among conditions in each experiments were examined by two-factor permutational multivariate analysis of variance (two-factor PERMANOVA; 999 permutations) using the `adonis2()` function in the R package `vegan`, with subsequent multiple pairwise comparisons conducted using the `pairwise.adonis2()` function. To identify indicator OTUs responsible for polyp bail-out, an LDA Effect Size (LEfSe) analysis was carried out for coral tissue libraries at the time of polyp bail-out in both experiments (24 h in the hypersaline experiment; 5-7 day in the hyperthermal experiment) using default settings. Differences were considered significant at $p < 0.05$ (adjusted p -values for the Tukey's HSD test and FDR-adjusted p -values for the multiple comparisons of beta diversity) for all statistical analyses.”

Line 183 & in Fig 2- indicate when polyp bail out occurred

Response: Thank you for your suggestion. We have added information regarding the timing of polyp bail-out both in the manuscript and in all figures except for the NMDS plots.

Revision: “When focusing on treatment effect at each time point, a discernible difference was found at onset of polyp bail-out (days 5-7) in the hyperthermal experiment. However, none of the pairwise comparisons in either experiment showed significant differences (FDR-adjusted p -value < 0.05).”

Discussion:

Line 262: results show a decrease in *Pseudomonas* in the treatment and an increase in the controls – it is not clear how these results indicate time dependent establishment and resilience.

Response: In the hyperthermal experiment we found that *Pseudomonas* increased in both the treatment and control groups throughout the cultivation (day 0 to days 5-7 in the control group and day 0 to day 2 in the treatment group), with the result that *Pseudomonas*-dominated communities in our corals. However, given that the aim of this study was to examine microbiome changes during bail-out induction, we think that the discussion about *Pseudomonas* changes may distract from the main purpose of the study. Therefore, we removed the related discussion in the revised manuscript. Nonetheless, we added text to the beginning of the second paragraph in the **Discussion** about time-dependent microbiome changes in our experiments.

Revision: “In this study, we observed CAB changes as a temporal response in both our experiments (especially the hyperthermal experiment). Given that culture conditions in our indoor aquaria (closed system; lower light intensity; artificial seawater) differed from those in which corals were acclimated before experiments (open system; natural sunlight; sand-filtered natural seawater), CAB changes during cultivation are not unexpected.”

Line 264: “elevated seawater temperatures seem to facilitate microbiome re-establishment in Pocillopora corals, as observed in our hyperthermal experiment.” What is the evidence for re-establishment?

Response: As mentioned in the previous response, we have now removed the related discussion in the revised manuscript.

Line 278: I disagree with this statement. Because the temperature and salinity were still being changed during the timing of polyp bail out, it is not clear if it is environmentally mediated microbial changes or host-induced microbial changes. Because no host-level data is shown, it is also difficult to understand the other physiological changes associated with increased temperature.

Response: Thank you for this comment. This sentence (and the whole paragraph) has been removed from the revised manuscript (as mentioned above). Some description here is now incorporated into the second paragraph of the **Discussion** regarding common microbiome changes in response to hypersaline or hyperthermal stress.

Revision: “Decreased *Endozoicomonas* and increases in *Rhodobacteraceae* or *Vibrio* have been frequently found in thermally stressed corals. However, these changes were not found in our corals, reflecting a species- or location-specific response in our *P. acuta*.”

Line 292: Opportunistic growth cannot be out ruled. For example, there are no water samples or environmental samples included, and there are site-specific differences in microbiomes, which may be why the taxa observed in this study were not observed in other studies. Right now these observations are correlative and not causative and should not be stated as causative.

Response: Thank you for your comment. We agree that opportunistic growth cannot be fully ruled out here. To accommodate the mentioned factors, the sentence has been revised.

Revision: “Common increases of these bacteria in both our experiments imply their involvement in development of polyp bail-out. Although, to our knowledge, no previous study has reported increases of these bacterial taxa in thermally or osmotically stressed corals, species- and site-specific differences in coral microbiomes may also contribute. The possibility of opportunistic growth of these bacteria thus cannot be fully excluded. Future studies of their capacity to induce polyp bail-out as pure isolates or a community may yield a clearer picture of their role in polyp bail-out.”

The discussion should include what microbial taxa are typically associated with *Pocillipora* in other studies and have been known to change with bleaching and salinity.

Response: Thank you for your suggestion. In the revised manuscript we have added related text to the first two paragraphs of the **Discussion**, including dominant bacteria commonly observed in *P. acuta* and changes commonly found in response to thermal or saline stress. Unfortunately, references about microbiome changes in response to hypersaline stress are scarce (only one was found).

Revision: “In both experiments, microbiomes in *P. acuta* corals were dominated by *Proteobacteria* (especially *Gammaproteobacteria* and *Alphaproteobacteria*), consistent with those identified in field-collected samples. At the OTU level, *Pseudomonas* sp. (OTU0001) and an unclassified *Rhodobacteraceae* bacterium (OTU0002) were the dominant bacterial taxa in non-bail-out libraries. *Pseudomonadaceae* and *Rhodobacteraceae* have predominated in *P. acuta* in Singapore and the Great Barrier Reef. Several marine isolates of *Pseudomonas* possess antimicrobial, antifungal, or biodegrading activity. Although evidence is yet not available for scleractinian corals, *Pseudomonas* bacteria isolated from the soft corals *Sarcophyton glaucum* and *Sinularia polydactyla* inhibit growth of some other bacteria and fungi and are thought to protect coral hosts against pathogens. Our results add further evidence to the association of *Pseudomonas* bacteria with *P. acuta*, warranting further investigation of its ecological functions to the coral holobionts. On the other hand, *Endozoicomonas* bacteria, another bacterial clade reportedly predominant in *P. acuta* and other *Pocillopora* corals, presented only sporadically in our coral samples. As microbiomes in *P. acuta* vary even within short geographic distances, inconsistency of our results with other studies is not surprising and can likely be attributed to site-specific variation in *P. acuta* microbiomes.

In this study, we observed CAB changes as a temporal response in both our experiments (especially the hyperthermal experiment). Given that culture conditions in our indoor aquaria (closed system; lower light intensity; artificial seawater) differed from those in which corals were acclimated before experiments (open system; natural sunlight; sand-filtered natural seawater), CAB changes during cultivation are not unexpected. In contrast to the time effect, the hypersaline treatment exerted no significant effect on microbiomes of *P. acuta*. Addition of hypersaline seawater in the treatment aquarium could have introduced exotic bacteria and represent another possible factor in the treatment effect (no water was added to the control aquarium). Nevertheless, our findings suggest a negligible effect from addition of water during 24 h. These results are also consistent with that reported in *F. granulosa*, in which microbiomes remained stable after short-term (4 h) exposure to hypersaline stress. Unfortunately, as studies about effects of hypersaline stresses on coral microbiomes are limited, a solid conclusion cannot be drawn without further investigation.”

Methods

How many tanks were used for each experiment?

Response: In this study, we used only one tank for each treatment in each experiment. This information has been further clarified in the **Materials and Methods**.

Revision: “One day prior to each experiment, small fragments (~1 cm branch tips; six fragments/colony for the hypersaline experiment and eight fragments/colony for the hyperthermal experiment) were separated from selected colonies using a clean bone cutter and placed in two 5-L indoor aquaria (one denoted as the treatment group and the other denoted as the control group).”

How many fragments were removed for sampling during each time point?

Response: At each time point, one fragment from each colony was sampled in both treatment and control groups. Total sampling numbers per time point are 5 fragments (from 5 colonies) from the treatment group, and 5 fragments from the control group. This information is now clarified in the **Materials and Methods**.

Revision: “From both treatment and control groups, one fragment from each colony was collected using a clean forceps at 0 h, 12 h, and 24 h (onset of polyp bail-out; defined as distinguishable colony dissociation and polyp detachment; Fig. 1), making a total of 5 fragments per condition per time point.”

“One fragment from each colony was collected using a clean forceps in both treatment (before temperature changes) and control groups at day 0, day 1, day 2, and when polyp bail-out was observed (days 5-7; Fig. 1), equaling 5 fragments per condition per time point.”

What types of heaters were used to raise the temperature and what type of thermometer was used? How temperature kept constant within the tank?

Response: Seawater temperatures in our experiments were controlled using two 110W aquarium heaters (products of Kotobuki Kogei, Japan) and were monitored with a ProfiLux 4 aquarium controller (GHL, Germany). This information is now provided in the **Materials and Methods**.

Revision: “Light was provided at 50 $\mu\text{mol photons/m}^2/\text{s}$ using a Mitras Lightbar 2 (GHL, Germany) with a 12-h light-dark cycle (light: 06:00-18:00) and seawater temperature was set at 25°C in both aquaria using two 110W aquarium heaters (Kotobuki Kogei, Japan). Seawater salinity and temperature were monitored with a ProfiLux 4 aquarium controller (GHL, Germany).”

How were coral samples taken for microbial analysis?

Response: At each sampling time point, coral fragments were collected using a clean forceps and immediately placed in sterile 1.5 mL Eppendorf tubes, which were then stored at -20°C. For samples of polyp bail-out, both the skeleton and detached polyps per colony were collected in the same Eppendorf tube as one sample. This information has been added to the **Materials and Methods**.

Revision: “Collected coral fragments were immediately placed in sterile 1.5 mL Eppendorf tubes and stored at -20°C. For each sample of polyp bail-out, detached polyps (collected using a micropipette and a sterile pipette tip with <100 μl seawater) and the remaining skeleton (with undetached polyps, if present; collected using a clean forceps) were collected as one sample to keep the sampling consistent during the experiment.”

What were the sampling and procedural controls for the 16S analysis?

Response: Negative controls were prepared from the DNA extraction step using only the tissue lysis buffer in the DNeasy Blood & Tissue Kit (QIAGEN, Japan) without coral tissues. This information has been added to the **Materials and Methods**.

Revision: “Two tissue-negative controls (containing only tissue lysis buffer) were included in the DNA extraction step.”

How many libraries were constructed? If more than one, how were the different libraries combined?

Response: In this study, one library was constructed from each sampled coral tissue. This means 5 libraries (from 5 coral colonies) * 2 groups (treatment and control) * 3 time points (0 h, 12 h, 24 h) = 30 libraries in the hypersaline experiment, and 5 libraries * 2 groups * 4 time points (0 day, 1 day, 2 day, 5-7 day) = 40 libraries in the hyperthermal experiment. Another two libraries were prepared from tissue-negative controls during the DNA extraction step (no coral tissue or seawater was added to the tissue lysis buffer). This information has been emphasized in the revised manuscript in the **Results and Materials and Methods**.

Revision:

(in the **Results**)

“A total of 16,760,948 reads were generated from 72 16S-rRNA gene libraries, including 30 libraries from the hypersaline experiment (5 colonies * 2 conditions * 3 time points), 40 libraries from the hyperthermal experiment (5 colonies * 2 conditions * 4 time points), and 2 tissue-negative controls included at the DNA extraction step.”

(in the **Materials and Methods**)

“In total, 70 coral tissue samples were collected, including 30 from the hypersaline experiment (5 colonies * 2 conditions * 3 time points) and 40 from the hyperthermal experiment (5 colonies * 2 conditions * 4 time points).”

Line 339: Not clear if the coral was sampled or the polyp that had bailed was sampled for 16S

Response: We are sorry for the confusion. For each sample of polyp bail-out, the mother fragment and bailed-out polyps were collected together as one sample. This information has been emphasized in the **Materials and Methods**.

Revision: “For each sample of polyp bail-out, detached polyps (collected using a micropipette and a sterile pipette tip with <100 µl seawater) and the remaining skeleton (with undetached polyps, if present; collected using a clean forceps) were collected as one sample to keep the sampling consistent during the experiment.”

Line 375: Why was the V5-V6 region used? The V4 region is usually used in coral microbiome studies.

Response: The V5-V6 region is also commonly used in coral studies. We chose this region based on the following references:

Damjanovic et al. (2020). Mixed-mode bacterial transmission in the common brooding coral *Pocillopora acuta*.

Röthig, et al. (2016). Long-term salinity tolerance is accompanied by major restructuring of the coral bacterial microbiome

Cárdenas et al (2022). Greater functional diversity and redundancy of coral endolithic microbiomes align with lower coral bleaching susceptibility

Related description is in the **Materials and Methods**.

Revision: “The primer pair specific for the V5-V6 region of the 16S-rRNA gene (784F: 5'-TCGTCGGCAGCGTCAGATGTGTATAAGAGACAGAGGATTAGATACCCTGGTA-3'; 1061R: 5'-GTCTCGTGGGCTCGGAGATGTGTATAAGAGACAGCRRACGAGCTGACGAC-3') was used to construct 16S-rRNA gene libraries, given their utility in detecting bacterial communities associated with corals (14, 69, 70).”

Data processing:

How was error and quality parameters decided?

Response: For 16S data processing, all parameters in Mothur were decided based on the default settings or the values suggested by the producer unless specified. This information has been added to the **Materials and Methods**.

Revision: “Given a median length of 308 bp in our assembled contigs, assembled contigs longer than 350 bp or shorter than 250 bp were filtered out using the ‘screen.seqs’ command with the parameters: minlength=250 and maxlength=350, assuming they represent incorrect/unsuccessful assembly. Criteria for ambiguous base and homopolymer filtering were set as suggested by the Mothur development team (maxambig=0, and maxhomop=8).”

When were paired end reads merged? Is that is what is meant as “assembling contigs” – which is typically not language used in 16S studies. Why was merging conducted before quality control of sequences?

Response: The ‘make.contigs’ step in Mothur assembled paired sequencing reads based on quality scores in the fastq files. This information is clarified in the **Materials and Methods**.

Revision: “We used Mothur version 1.44.1 to analyze sequencing data. Raw data were first assembled into contigs using the ‘make.contigs’ command in Mothur with default parameter settings, which assembles paired sequencing reads into contigs according to the corresponding quality scores in fastq files.”

Statistics

It seems as though the question the authors are asking is how did the microbiome of corals change before and after polyp bail out – this does not seem to be what was tested statistically, or it is not clear what was tested statistically to demonstrate that effect. Models should include Treatment, Time, before/after bail out, and an effect for the coral sampled (or colony sampled).

Response: Thank you for these comments. In this study we sought to understand microbiome dynamics during induction of polyp bail-out, as we expected to see gradual changes of microbiomes during our experiments (although the results actually point to more of a sudden change at the onset of bail-out). Statistical analysis was therefore conducted to examine differences between the treatment and control groups at each time point during our

experiments. We have re-analyzed the datasets using two-way ANOVA (for alpha diversity) and two-factor PERMANOVA (for beta diversity) to check the effects of treatment and time, and their interaction. Unfortunately, as the coral samples for each condition were composed of 5 fragments from 5 colonies (one per colony), the effects of individual variation cannot be tested. These modifications have been updated in a new subsection (**Statistical analyses**) in the **Materials and Methods**.

Revision: “Statistical analyses for alpha and beta diversity were conducted in the R environment (v3.6.1). Two-way analysis of variance (two-way ANOVA) was conducted using the function `aov()` to examine statistical differences in alpha diversity among conditions (treatments and time points) for each experiment, followed by a Tukey's honestly significant difference (HSD) test for post-hoc analysis. Statistic differences in bacterial composition among conditions in each experiments were examined by two-factor permutational multivariate analysis of variance (two-factor PERMANOVA; 999 permutations) using the `adonis2()` function in the R package `vegan`, with subsequent multiple pairwise comparisons conducted using the `pairwise.adonis2()` function. To identify indicator OTUs responsible for polyp bail-out, an LDA Effect Size (LEfSe) analysis was carried out for coral tissue libraries at the time of polyp bail-out in both experiments (24 h in the hypersaline experiment; 5-7 day in the hyperthermal experiment) using default settings. Differences were considered significant at $p < 0.05$ (adjusted p -values for the Tukey's HSD test and FDR-adjusted p -values for the multiple comparisons of beta diversity) for all statistical analyses.”

Line 403: comparison across libraries is not clear statement is not clear. What treatments/days were compared for the PERMANOVA? How were multiple comparisons of the data taken into account? What programs were used to perform the PERMANOVA?

Response: In the revised manuscript, we re-analyzed these datasets with two-factor PERMANOVA (using `adonis2()` in R). At both class and OTU levels, the results showed a significant interaction between treatment and time in only the hyperthermal experiment. Multiple comparisons (using `pairwise.adonis2()` in R) were therefore performed for the hyperthermal experiment. The resulting p -values were FDR-adjusted to account for multiple comparisons. Since this study sought to examine microbiome dynamics caused by bail-out induction treatments, we only reported comparisons of treatment vs. control at individual time points in the revised manuscript. Related description is now provided in the **Results** and **Materials and Methods**.

Revision:

(in the **Results Class level**)

“When focusing on treatment effect at each time point, a discernible difference was found at onset of polyp bail-out (days 5-7) in the hyperthermal experiment. However, none of the pairwise comparisons in either experiment showed significant differences (FDR-adjusted p -value < 0.05).”

(in the **Results OTU level**)

“When focusing on the treatment effect at individual time points, differences between treatment and control groups were significant along the whole hyperthermal experiment, except on day 1.”

(in the **Materials and Methods**)

“Statistical analyses for alpha and beta diversity were conducted in the R environment (v3.6.1). Two-way analysis of variance (two-way ANOVA) was conducted using the function `aov()` to examine statistical differences in alpha diversity among conditions (treatments and time points) for each experiment, followed by a Tukey's honestly significant

difference (HSD) test for post-hoc analysis. Statistic differences in bacterial composition among conditions in each experiments were examined by two-factor permutational multivariate analysis of variance (two-factor PERMANOVA; 999 permutations) using the `adonis2()` function in the R package `vegan`, with subsequent multiple pairwise comparisons conducted using the `pairwise.adonis2()` function. To identify indicator OTUs responsible for polyp bail-out, an LDA Effect Size (LEfSe) analysis was carried out for coral tissue libraries at the time of polyp bail-out in both experiments (24 h in the hypersaline experiment; 5-7 day in the hyperthermal experiment) using default settings. Differences were considered significant at $p < 0.05$ (adjusted p -values for the Tukey's HSD test and FDR-adjusted p -values for the multiple comparisons of beta diversity) for all statistical analyses."

Line 405: What does the `metastats` command do in `mothur`? Specify what this analysis is, what the assumptions are and how it is run.

Response: Since we observed prominent microbiome changes in both experiments at the time of polyp bail-out, we used `metastats` (a built-in function in `mothur` that performs a non-parametric t-test) to determine significantly differentiated OTUs between treatment and control groups when bail-out occurred (24 h in the hypersaline experiment and days 5-7 in the hyperthermal experiment). However, this method is not commonly used in microbiome studies. Therefore, we re-analyzed the data using LDA Effect Size (LEfSe) (Segata *et. al*, 2010), a more commonly used method for identifying indicator OTUs that characterize differences between compared conditions. According to the description in the corresponding website (<http://huttenhower.sph.harvard.edu/galaxy/>), LEfSe "*first use the non-parametric factorial Kruskal-Wallis sum-rank test to detect features with significant differential abundance with respect to the class of interest; biological significance is subsequently investigated using a set of pairwise tests among subclasses using the (unpaired) Wilcoxon rank-sum test. As a last step, LEfSe uses Linear Discriminant Analysis to estimate the effect size of each differentially abundant feature and, if desired by the investigator, to perform dimension reduction.*" In the analysis, we put the control and treatment groups at the time of polyp bail-out (24 h in the hypersaline experiment and days 5-7 in the hyperthermal experiment) as the "class" and colony ID as the "subclass" to allow paired Wilcoxon rank-sum test. Alpha values for both the Kruskal-Wallis and Wilcoxon tests were set at 0.01 to accommodate the small sample size in our dataset (N=4). The default threshold of LDA score was set at 2.0. This modification has been updated in the **Materials and Methods**.

Revision: "To identify indicator OTUs responsible for polyp bail-out, an LDA Effect Size (LEfSe) analysis was carried out for coral tissue libraries at the time of polyp bail-out in both experiments (24 h in the hypersaline experiment; 5-7 day in the hyperthermal experiment) using default settings."

Figures/Tables

Table 1: Does >1% refer to the entire dataset? Are these in the top 1% of either experiment or across both experiments (i.e., when the data are combined).

Response: In the original Table 1, the 1% parameter referred to the entire dataset (data from both experiments were combined). In the revised manuscript we decided to include generic names in the figures (instead of just showing OTU numbers); therefore, the information in Table 1 became redundant, so we removed this table from the revised manuscript.

Figures should include when polyp bail out occurred to understand the comparisons made.

Response: Thank you for this suggestion. We have now highlighted timing of polyp bail-out in all figures.

Revision:

Why are both Figures 2 and Figure 2 shown? The taxa information could be included in figure 3 and grouped by class and that would provide the same information.

Response: Thank you for this suggestion. We tried to combine Figures 2 and 4 into one figure. However, showing class-level taxonomy with OTUs increases the complexity of the figure (too many colors would be required to represent OTUs and classes), which defeats the purpose of the figure. A combined figure also unavoidable causes smaller text, especially after we decided to use generic names instead of OTU numbers in Figure 4. Considering these issues, we decided to keep both figures in the revised manuscript.

Figure 4 and 5: Indicate taxonomic information (e.g. Genus) with OTUs in figures 4 and 5

Response: Thank you for your suggestion. Taxonomic information (at the generic level) has now been included after OTU numbers in Figures 5 and 6.

Revision:

Fig. 5

Fig. 6

Figure 6: indicate the number of sample, and how many corals exhibited polyp bail out at each time point.

Response: Thank you for this suggestion. We have now incorporated the required information in the figure (the figure has been renumbered as Figure 1 in the revised manuscript, based on Reviewer 2's suggestion).

Revision:

Fig. 1

The authors of this manuscript sought to characterize members of the microbiome that may be responsible for polyp bailout. While they did not empirically test candidate microbes that were identified in their study and their role in polyp bailout, it is a novel and important study that can begin to test this hypothesis. Overall I believe the study is effective in its approach and they successfully identified microbes that are associated with polyp bailout in *Pocillopora* spp. However, I have several major and minor concerns with the methodology that should be addressed.

Overall, there needs to be more detail in the methods portion of this manuscript. I have outlined my concerns below, in no particular order of importance:

1. Line 326 - is 50 umol of light used in the indoor aquaria comparable to the outdoor light settings? It does not appear likely. Please inform the reader how light could potentially alter the microbiome (if at all).

Response: Thank you for calling attention to the effect of light. Unfortunately, we couldn't find any references discussing effects of light intensity changes on coral microbiomes. Moreover, our indoor aquaria differed from outdoor aquaria in regard to multiple factors, including light intensity, seawater source, and open vs. closed systems. We have added discussion of these factors in the second paragraph of the **Discussion**.

Revision: "In this study, we observed CAB changes as a temporal response in both our experiments (especially the hyperthermal experiment). Given that culture conditions in our indoor aquaria (closed system; lower light intensity; artificial seawater) differed from those in which corals were acclimated before experiments (open system; natural sunlight; sand-filtered natural seawater), CAB changes during cultivation are not unexpected."

2. Of the corals used in this study, 1 out of 5 of the coral was *P. damicornis*. It is unclear if the microbiome of *P. damicornis* is comparable to that of *P. acuta* as they are entirely different species. I believe that the microbial dataset for *P. damicornis* should be removed as it could be a confounding variable, or that it should be justified why it was kept in the study and the figures should clearly denote which sample came from *P. damicornis* as it is not obvious which one it may be, especially as some coral microbiomes appear to behave differently to others. For example, the microbial community from the fifth bar in Figure 4 has a dramatically different abundance of OTU0015 compared to the others - is that the *P. damicornis* sample?

Response: Thank you for your suggestions. We have now excluded data from *P. damicornis* in our analyses. The description has been updated in the **Results** in the revised manuscript. All updated figures now include only *P. acuta* (N=4). Results including *P. damicornis* are now presented in supplementary Figures 3-6. In supplementary Figures 4 and 6 (bacterial composition at class and OTU levels, respectively), data of *P. damicornis* are placed in the fifth bar in each condition (description provided in figure legends).

Revision: "Given that *Pocillopora* #15 was identified as *P. damicornis* and showed discernible physiological response differences (bleaching before bail-out) compared to the others, the corresponding datasets were not included in subsequent analyses (analyses including *Pocillopora* #15 are provided in Fig. S3-6)."

3. Were the corals kept in aquaria with no flow or filter for 7 days during your heat-stress experiment? Please clarify the aquarium setup.

Response: In both experiments, both treatment and control tanks were equipped with a slim filter pump for water circulation. This information is now added to the **Materials and Methods**.

Revision: “Each aquarium was equipped with a 3.4 W slim filter pump (GEX, Japan) for water circulation.”

4. Line 333 - Water was added to the hypersaline treatment to slowly increase the salinity, however the control treatment had no additional seawater added. This is a confounding variable and should be justified.

Response: As the experiment for hypersaline treatment was conducted only for 24 h, the salinity fluctuation in the control tank was assumed to be negligible (<1‰). We therefore didn't add artificial seawater or fresh water to the control group during the hypersaline experiment. However, this might have introduced foreign bacteria to the treatment tank, causing an additional deviation of the treatment group from the control group. The same issue also applies to the hyperthermal experiment, in which higher volumes of fresh RO water were added to compensate for greater evaporation in the treatment tank. These factors are now discussed in the **Discussion**.

Revision:

(for the hypersaline experiment)

“Addition of hypersaline seawater in the treatment aquarium could have introduced exotic bacteria and represent another possible factor in the treatment effect (no water was added to the control aquarium). Nevertheless, our findings suggest a negligible effect from addition of water during 24 h.”

(for the hyperthermal experiment)

“Another potential effect from the hyperthermal treatment is stronger evaporation at elevated temperatures, which caused higher salinity fluctuations in the treatment aquarium. Meanwhile, to compensate for stronger evaporation, higher volumes of fresh water were added to the treatment aquarium, representing another possible factor in differences between the treatment and control groups in our hyperthermal experiment.”

5. Line 333 - In addition, please add information on what the source of your artificial seawater is (what salt manufacturer) and whether it was sterilized with a filter and/or autoclaving.

Response: In this study we prepared artificial seawater by dissolving artificial sea salt (Kaisuimaren, Japan) in reverse osmosis (RO) water. No additional filtering or autoclaving were conducted. This information is now provided in the **Materials and Methods**.

Revision: “Seawater was prepared at 35‰ by dissolving artificial sea salt (Kaisuimaren, Japan) in reverse osmosis (RO) water.”

6. Line 350 - What is the source of the freshwater in the heat-stress experiment? Tap water? DI water? Was this sterilized as well? Was the salinity fluctuation significant?

Response: In the hyperthermal experiment, we used RO water as the source of freshwater to compensate for evaporation-driven salinity changes. During the experiment, salinity fluctuation in the control group (at 25°C) was <1 ‰. In the treatment group, however, due to higher temperatures, salinity increases ranged from ~1‰ (at 28‰) to ~3‰ (at 34°C). This information has been added to the **Materials and Methods** and the **Discussion**.

Revision:

(Materials and methods)

“To compensate for water evaporation during the experiment (salinity fluctuation <1% in the control group and <3% in the treatment group at elevated temperatures), fresh RO water was added to both aquaria every day (after tissue sampling) to a fixed water level.”

(Discussion)

“Another potential effect from the hyperthermal treatment is stronger evaporation at elevated temperatures, which caused higher salinity fluctuations in the treatment aquarium. Meanwhile, to compensate for stronger evaporation, higher volumes of fresh water were added to the treatment aquarium, representing another possible factor in differences between the treatment and control groups in our hyperthermal experiment. It is also worth noting that bacterial communities showed a significant difference between treatment and control groups even at the beginning of our hyperthermal experiment (day 0). The significant effect of hyperthermal treatment in this study therefore may also be attributed to a cage effect in our experimental setup. As bacterial composition in seawater was not examined in this study (due to the small size and limited number of aquaria), how water addition and cage effects influenced coral microbiomes remains unclear. Nevertheless, given that bacterial communities in treatment and control groups were not significantly different at the class level (per time point), observed variations at the OTU level probably reflect a shift between functionally redundant bacterial groups without significant detriment to coral physiology, as suggested by H.-P. Lu et al. (55) and A. Hernandez-Agreda et al. (56).”

7. Line 359 - Please clarify what was used in the negative control for the PCR. Nuclease-free water from the kit? Or seawater from your seawater source?

Response: In this study, negative controls were prepared by running the DNA extraction procedure without adding a sample to the tissue lysis buffer of the DNA extraction kit (no coral tissue, seawater, or nuclease-free water). This information has been now added to the **Materials and Methods**.

Revision: “Two tissue-negative controls (containing only tissue lysis buffer) were included in the DNA extraction step.”

8. An important concern: It appears that only two 5L aquaria were used for each of the hypersaline and heat stress experiments. This could cause significant cage effects on the microbial community (especially from coprophagy) and should be clearly justified and clarified in the discussion section. If this was not the case, and each coral

fragment was split into their own respective aquaria (for a total of 30 aquariums for hypersaline and 40 aquariums for heat stress), please clarify.

Response: Yes, your understanding is correct. In this study we used only two aquaria for each experiment (one for the treatment and the other for the control group). We designed these experiments to avoid inter-tank variation regarding the rate of salinity increase (in the hypersaline experiment) and seawater heating rate (in the hyperthermal experiment). Cage effects therefore cannot be neutralized. We have now mentioned this factor in the **Discussion**.

Revision: “It is also worth noting that bacterial communities showed a significant difference between treatment and control groups even at the beginning of our hyperthermal experiment (day 0). The significant effect of hyperthermal treatment in this study therefore may also be attributed to a cage effect in our experimental setup. As bacterial composition in seawater was not examined in this study (due to the small size and limited number of aquaria), how water addition and cage effects influenced coral microbiomes remains unclear.”

9. How is data from days 5-7 of the heat stress experiment compiled? Are the coral only sampled after polyp bailout? Or were all genets sampled at both day 5 and day 7? On all figures it appears that days 5-7 were somehow integrated together and it is unclear how that was done.

Response: In the hyperthermal experiment, 2 of the coral fragments showed a bail-out response on day 5, while the other 3 did so on day 7. As we sought to examine changes in the coral microbiome during development of polyp bail-out, these samples were grouped to represent the condition at the onset of polyp bail-out. This information is now emphasized in the **Results**.

Revision: “Unlike the hypersaline experiment, polyp bail-out in the hyperthermal experiment was observed at two different times: at day 5 in two of the fragments (34°C; *Pocillopora* samples #15 and #24) and at day 7 in the remaining three fragments (34°C; *Pocillopora* samples #10, #22, and #23; Fig. 1). As both represented onset of polyp bail-out, samples at day 5 and day 7 were pooled to represent conditions of polyp bail-out onset (denoted days 5-7) in subsequent analyses (so too, corresponding samples in the control group).”

10. All OTUs present in the negative controls should be removed from your analysis. Especially OTU0001 as the abundance of reads in the negative controls were in the same magnitude as the coral samples. Fortunately the other OTUs of interest (0004, 0012, and 0039) do not appear to be present in the negative control so it should not detract from the overall conclusion of the manuscript. If the authors choose to keep OTU0001 in their analysis, it should be justified, and included in Figure 5 as well. Please see <https://doi.org/10.3389/fmicb.2022.1007877> for best practices on treating PCR contamination data.

Response: Thank you for your comments. The identified taxonomy of OTU0001 (*Pseudomonas*) is common to a wide range of samples, including corals and PCR reagents, causing overlap of contaminant ASVs/OTUs with true coral-associated bacterial ASVs/OTUs. The same situation also applies to other OTUs, including OTU0004, but to a lesser degree (several thousands of reads in coral tissue samples vs. <3 reads in negative controls). Given the short length of our target gene fragment

(~300 bp), it is difficult to clearly distinguish contaminant ASVs/OTUs from true ASVs/OTUs in coral tissue samples. Complete removal of OTUs present in negative controls will lose considerable information and may not reflect the true microbiomes of our corals. Considering that contamination should be proportional to the concentration of the contaminants (related to true samples), we decided to use the software microDecon for decontamination, which conducts de-contamination using a proportion-based approach. This process removed about 15% of the reads from OTU0001 (2,697,496 reads -> 2,319,680 reads among all coral tissue libraries), although in most samples it is still the dominant OTU. Related information has been added to the **Materials and Methods**.

Revision: “De-contamination was then conducted using microDecon, software that assumes fixed ratios among contaminant contigs in blank (tissue-negative controls in this study) and true samples and removes them from true samples using a proportion-based approach. Parameters in microDecon were set as the default, except for ‘thresh’, which was set to 1 to accommodate genotype-specific contigs.”

11. How did you ensure that equal sampling of the coral skeleton and polyps occurred in the polyp bailout and the non-polyp bailed out samples occur? Was the entire 1cm fragment processed? This portion needs more details, as the differences in microbiomes could potentially be explained by sampling differences post-bailout (for example, if your OTUs of interest are over-represented in the polyp compared to the skeleton).

Response: For samples of polyp bail-out, all detached polyps and the remaining skeleton were collected together (in the same Eppendorf tube) to ensure equal sampling of non-bail-out samples. The sampling process has been further clarified in the **Materials and Methods**.

Revision: “From both treatment and control groups, one fragment from each colony was collected using a clean forceps at 0 h, 12 h, and 24 h (onset of polyp bail-out; defined as distinguishable colony dissociation and polyp detachment; Fig. 1), making a total of 5 fragments per condition per time point. Collected coral fragments were immediately placed in sterile 1.5 mL Eppendorf tubes and stored at -20°C. For each sample of polyp bail-out, detached polyps (collected using a micropipette and a sterile pipette tip with <100 µl seawater) and the remaining skeleton (with undetached polyps, if present; collected using a clean forceps) were collected as one sample to keep the sampling consistent during the experiment.”

I also have several minor concerns:

1. Lines 82-95 seem too speculative and do not contribute to the overall introduction of polyp bailout.

Response: Thank you for your suggestion. We have removed this paragraph in the revised manuscript.

2. Figure 6 should be the first figure in the manuscript and visually denote when polyp bailout occurs. Figure 1 can be brought up in Line 142.

Response: Thank you for your suggestion. Figure 6 has become Figure 1 in the revised manuscript (cited in the subsection “*Coral polyp bail-out and species identification*” under the **Results**).

3. Figures 1-4, visually labeling the individual coral genets and being consistent with the placement of the genet would be helpful. Also, visually annotating when polyp bailout occurs for each genet would be immensely helpful. And please change “treatment” to “hypersaline” and “hyperthermal” for clarity (like it is done in Figure 1)

Response: Thank you for your suggestions. In the revised figures we have removed data of *P. damicornis* (#15). Results including *P. damicornis* are provided in Supplementary Figures 3-6. In revised Figures 3 and 5, coral colonies are now placed in the same order for all conditions (#10, #22, #23, #24) and described in figure legends. Timing of polyp bail-out is also highlighted in the figures. The annotation of “Treatment” has also been changed to hypersaline or hyperthermal, as suggested.

Revision:

Fig. 3

Fig. 5

4. Line 185 - Two p-values are listed but both are labeled as hypersaline.

Response: Thank you for pointing out this typo. The second label should be hyperthermal and it has been fixed in the revised manuscript.

5. Figure 3 - please remove the “_” from control and treatment labels.

Response: Thank you for the suggestion. All “_” have now been removed from labels.

Revision:

Fig. 4

6. Figure 3 - this dataset illustrates the Anna Karenina principle well and should be included in the discussion that could also potentially explain polyp bailout (see: <https://doi.org/10.1038/nmicrobiol.2017.121>)

Response: Thank you for your suggestion. A brief discussion about the Anna Karenina effect has been added to the **Discussion**.

Revision: “In the NMDS plots, higher dispersion was found among libraries of polyp bail-out compared to the corresponding control groups, but was less clear before onset of bail-out. This suggests a possible “Anna Karenina” effect during development of polyp bail-out.”

7. Figure 5 legend requires information on statistical tests and what the asterisk denotes.

Response: Thank you for your suggestions. In the revised manuscript we changed from metastat (in Mothur) to LEfSe, as the method for identifying indicator OTUs of polyp bail-out. No additional statistical test was performed to examine abundance differences between treatment and control groups (per time point) for each indicator OTU. Asterisks have therefore been removed from the figure.

May 14, 2023

Dr. Po-Shun Chuang
Biodiversity Research Center Academia Sinica
Taipei
Taiwan

Re: Spectrum00257-23R1 (Bacterial community shifts during polyp bail-out induction in *Pocillopora* corals)

Dear Dr. Po-Shun Chuang:

Dear Authors,

Please see reviewer #1 comments. Please address these issues.

Thank you!

Link Not Available

Sincerely,

Nikki Traylor-Knowles

Journals Department
Reviewer comments:

Reviewer #1 (Comments for the Author):

Spectrum Review:

The authors added substantial revisions to this manuscript to prepare it for review. Although this is an exciting question, I take issue with the experimental design, which is not truly replicated (only one tank was used per treatment level), indicating pseudoreplication- so all samples were not independent of each other, and were repeatedly sampled. This drawback was not mentioned, nor was the fact that the samples were not independent, which makes parametric statistical tests not correct to do.

Line 326:

Experimental design is pseudoreplicated (only 1 tank is used in which treatment is applied) and replicates are non independent. Analyses conducted do not reflect this flawed design.

Line 181: NMDS is not an analysis, it is a visualization tool, and there was no significant separation in the hypersaline treatment based on the PERMANOVA

Line 273: If dispersion is a major result, an analysis of dispersion should be conducted, it was not.

Line 294: Visuals provided do not preclude bleaching or other health effects of the treatment, which should be discussed.

Figure 1: Images do not clearly indicate polyp bail out and appear as though tissue sloughing off of corals, indicating death. Please highlight the bail out polyps and images of bailed out polyps, this may also require further clarification.

Figure 3: Y axis label missing

Figure 5: Color for OTU001 and Others is too similar. No Y axis label

Figure 6: Labels are backwards, legend should indicate what box and whiskers indicate.

Figures 3 and 5: only one is necessary

Reviewer #2 (Comments for the Author):

The authors have addressed all of my concerns. I do believe that the cage effect limits the interpretation of the data heavily, as it is a significant confounding variable. However, I commend the authors for not obscuring the possibility of this effect in their manuscript.

Staff Comments:

Preparing Revision Guidelines

Please return the manuscript within 60 days; if you cannot complete the modification within this time period, please contact me. If you do not wish to modify the manuscript and prefer to submit it to another journal, please notify me of your decision immediately so that the manuscript may be formally withdrawn from consideration by Microbiology Spectrum.

Corresponding authors may join or renew ASM membership to obtain discounts on publication fees. Need to upgrade your

membership level? Please contact Customer Service at Service@asmusa.org.

Reviewer #1 (Comments for the Author):

Spectrum Review:

The authors added substantial revisions to this manuscript to prepare it for review. Although this is an exciting question, I take issue with the experimental design, which is not truly replicated (only one tank was used per treatment level), indicating pseudoreplication- so all samples were not independent of each other, and were repeatedly sampled. This drawback was not mentioned, nor was the fact that the samples were not independent, which makes parametric statistical tests not correct to do.

Response: Thank you for your comments. We have now mentioned the limitations of this study in the **Discussion**. Statistical analyses of alpha diversity have also been changed from two-way ANOVA and Tukey's HSD test to the corresponding non-parametric methods (Friedman's rank sum test and pairwise Wilcoxon rank sum test). Appropriate sections of the manuscript have been updated as indicated below.

Revision:

In Results

“In the hypersaline experiment, a significant difference was found among time points for the Chao1 index (Friedman's rank sum test; $p < 0.05$; Fig. 2), whereas in the hyperthermal experiment, significant differences were identified among time points for the Shannon and inverse Simpson indexes. Neither experiment showed a significant treatment effect. Although for all three indexes higher values were found at the onset of polyp bail-out in both experiments (24 h and days 5-7 in the hypersaline and hyperthermal experiments, respectively), none of the pairwise comparisons to the corresponding control groups showed significant differences after FDR adjustment (pairwise Wilcoxon rank sum test). Detailed results of statistical analyses are provided in Table S1 and a corresponding plotting including *Pocillopora* #15 is provided in Fig. S3.”

In Discussion

“It should also be mentioned that due to the design of this study (limited size of samples, only one aquarium for each treatment group, etc.), the present results should be interpreted with caution. These findings, however, provide a basis to test the hypothetical involvement of microbes in polyp bail-out.”

In Materials and Methods

“Given that samples collected at each time point in this study represented only pseudo-replicates (only one aquarium was used for each treatment in our experiments) and were repeatedly sampled from the same colonies, non-parametric statistical analyses were applied. Friedman's rank sum test was conducted to examine statistical differences in alpha diversity among conditions (treatments or time points) for each experiment. Although Friedman's test doesn't test for interactions between factors, post-hoc analysis for alpha diversity was conducted using the pairwise Wilcoxon rank sum test for all possible combinations of conditions (treatments \times time points) to allow examination of differences between treatment and control groups at specific time points.”

Line 326: Experimental design is pseudoreplicated (only 1 tank is used in which treatment is applied) and replicates are non independent. Analyses conducted do not reflect this flawed design.

Response: Thank you for raising this issue. We admit that our sampling in this study represented only pseudo-replication. To accommodate this issue, we have changed analytical methods for alpha diversity to non-parametric methods (from 2-way ANOVA + Tukey's HSD test to Friedman's rank sum test + pairwise Wilcoxon rank sum test). PERMANOVA and LefSe are unchanged, as they use non-parametric methods. This issue has been further clarified in the **Materials and Methods** and **Discussion** as limitations of this study. Corresponding revision has also been made to the **Results**.

Revision:

In Results

“In the hypersaline experiment, a significant difference was found among time points for the Chao1 index (Friedman's rank sum test; $p < 0.05$; Fig. 2), whereas in the hyperthermal experiment, significant differences were identified among time points for the Shannon and inverse Simpson indexes. Neither experiment showed a significant treatment effect. Although for all three indexes higher values were found at the onset of polyp bail-out in both experiments (24 h and days 5-7 in the hypersaline and hyperthermal experiments, respectively), none of the pairwise comparisons to the corresponding control groups showed significant differences after FDR adjustment (pairwise Wilcoxon rank sum test). Detailed results of statistical analyses are provided in Table S1 and a corresponding plotting including *Pocillopora* #15 is provided in Fig. S3.”

In Discussion

“It should also be mentioned that due to the design of this study (limited size of samples, only one aquarium for each treatment group, etc.), the present results should be interpreted with caution. These findings, however, provide a basis to test the hypothetical involvement of microbes in polyp bail-out.”

In Materials and Methods

“Given that samples collected at each time point in this study represented only pseudo-replicates (only one aquarium was used for each treatment in our experiments) and were repeatedly sampled from the same colonies, non-parametric statistical analyses were applied. Friedman's rank sum test was conducted to examine statistical differences in alpha diversity among conditions (treatments or time points) for each experiment. Although Friedman's test doesn't test for interactions between factors, post-hoc analysis for alpha diversity was conducted using the pairwise Wilcoxon rank sum test for all possible combinations of conditions (treatments \times time points) to allow examination of differences between treatment and control groups at specific time points.”

Line 181: NMDS is not an analysis, it is a visualization tool, and there was no significant separation in the hypersaline treatment based on the PERMANOVA.

Response: Thank you for your correction. We have changed “analysis” to “plotting” in this sentence and moved it to a later position to make the paragraph more readable.

Revision: “Consistent with the statistical analysis, non-metric multidimensional scaling (NMDS) plotting showed clear separation of treatment and control groups at the onset of polyp bail-out in the hyperthermal experiment (day 5-7), while the differentiation was less clear in the hypersaline experiment (Fig. 4; results including *Pocillopora* #15 are provided in Fig. S7).”

Line 273: If dispersion is a major result, an analysis of dispersion should be conducted, it was not.

Response: Thank you for your comment. The dispersion was analyzed by homogeneity of molecular variance analysis (HOMOVA). Although greater variances were found among libraries of polyp bail-out in both experiments compared to their counterparts, the differences were not significant after FDR adjustment (non-adjusted p -value was significant for the comparison in the hyperthermal experiment). For this reason, we have removed this discussion from the manuscript. A corresponding description of HOMOVA has been added to the **Results** and the **Materials and Methods**.

Revision:

In **Results**

“Homogeneity of molecular variance analysis (HOMOVA) identified greater variances at the onset of polyp bail-out in both experiments. However, the differences were not significant after FDR adjustment (Table S4).”

In **Discussion**

Removal of the corresponding discussion.

In **Materials and Methods**

“Differences in data variance between conditions (treatments \times time points) in each experiment were examined by homogeneity of molecular variance analysis (HOMOVA; 1,000 permutations), conducted in Mothur.”

Line 294: Visuals provided do not preclude bleaching or other health effects of the treatment, which should be discussed.

Response: Thank you for your comment. In the previous revision we removed the bleached sample from analysis and placed all data that include the bleached sample in supplementary figures. To make the **Discussion** more readable, we have added figure citations wherever applicable in the **Discussion**. We have also added further description about bailed-out polyp morphology to the **Results** and the **Discussion** about possible health effects of hyperthermal treatment on coral polyps.

Revision:

In **Results**

“Detached polyps in both experiments showed intact morphological features similar to those reported in earlier studies (31, 33), including round to cylindrical bodies and intact tentacles, indicating successful induction of polyp bail-out instead of tissue sloughing or polyp death.”

In **Discussion**

“Notably, in *P. damicornis* (*Pocillopora* #15) we observed bleaching at day 3 in the hyperthermal experiment, prior to the occurrence of polyp bail-out (Fig. S1). However, bleaching was not observed in our *P. acuta* corals (Fig. 1), suggesting different resistance to bleaching in the two coral species. In an earlier study, co-occurrence of bleaching and polyp bail-out was observed in *P. damicornis* under hyperthermal stress (62). Interestingly, bailed-out polyps in the same study showed less clear morphological differentiation, which was also observed in some of our bleached, bailed-out polyps from *P. damicornis*. This un-differentiated polyp morphology resembles to that of degenerated polyps reported in Chuang et al. (21), implying possible physiological damage in bailed-out polyps by hyperthermal

stress. Given that polyp recovery after bail-out is likely an energetically demanding process, thermal bleaching must significantly reduce recovery of bailed-out polyps, compromising their resettlement capacity. The relative developmental speeds of bail-out and bleaching, therefore, may have strong implications for dispersal potential of corals against the specter of future climate change.”

Figure 1: Images do not clearly indicate polyp bail out and appear as though tissue sloughing off of corals, indicating death. Please highlight the bail out polyps and images of bailed out polyps, this may also require further clarification.

Response: Thank you for your suggestions. In this study, we used bail-out induction methods published in earlier studies, which showed viability of coral polyps after the induced bail-out response (Chuang & Satoshi, 2020, *Coral Reefs*; Gösser et al., 2021, *Coral Reefs*). Although in this study we didn’t monitor survivorship of bailed-out polyps after the response (all bailed-out polyps were collected for 16S rRNA gene library construction), morphology of bailed-out polyps was similar to those reported in earlier studies. We believe the response we observed in this study is bail-out, rather than death. We have updated **Figure 1** by including photos of bailed-out polyps. Related description has also been added to the **Results**.

Revision:

In Results

“Detached polyps in both experiments showed intact morphological features similar to those reported in earlier studies (31, 33), including round to cylindrical bodies and intact tentacles, indicating successful induction of polyp bail-out instead of tissue sloughing or polyp death.”

Updated **Figure 1** and

Figure S1

Figure 3: Y axis label missing

Response: Thank you for raising this issue. We have added a Y-axis label to the figure. This figure has also been moved to supplementary figure S4.

Updated **Figure S4**

Figure 5: Color for OTU001 and Others is too similar. No Y axis label

Response: Thank you for raising this issue. We have added Y-axis labels to Fig. 3 (moved to supplementary figure S4) and Figure 5 (changed to Figure 3) and changed the color code of “Others” from blue to black in both figures to make them more distinguishable.

Updated **Figure 3**

Figure 6: Labels are backwards, legend should indicate what box and whiskers indicate.

Response: Thank you for your comment. The boxes and whiskers in this figure indicate the quartiles and full range of the datasets. We have added this information and reversed the Y labels in the legend. This figure has also been changed to Figure 5.

Revision:

In Figure 5 legend

“Abundance changes of OTU0004 (*Thalassospira* sp.), OTU0012 (*Marisediminitalea* sp.), OTU0036 (*Rhodobacteraceae*_unclassified), and OTU0080 (*Myxococcales*_unclassified) in both experiments. Boxes and whiskers indicate the quartiles and full range of the datasets, respectively.”

Updated Figure 5

Figures 3 and 5: only one is necessary

Response: Thank you for this comment. We have moved Figure 3 to supplementary figure S4. Corresponding description has been updated in the manuscript.

Revision:

In Results

“When grouping OTUs according to their classes, all non-bail-out *P. acuta* libraries were dominated by *Gammaproteobacteria* (mean \pm standard deviation = $52 \pm 13\%$) and *Alphaproteobacteria* ($33 \pm 12\%$; Fig. S4; results including *Pocillopora* #15 are provided in Fig. S5).”

Reviewer #2 (Comments for the Author):

The authors have addressed all of my concerns. I do believe that the cage effect limits the interpretation of the data heavily, as it is a significant confounding variable. However, I commend the authors for not obscuring the possibility of this effect in their manuscript.

Response: Thank you for your comments in the previous round of peer review. We appreciate your comment about the Anna Karenina effect in our dataset. However, given that the difference in variances between treatment and control groups (in both experiments) were not statistically significant, we decided to remove the relevant discussion. Nevertheless, how the bacterial OTUs that are potentially responsible for polyp bail-out (such as OTU0004 in our study) are correlated with the Anna Karenina effect would be an interesting topic for future studies.

June 8, 2023

Dr. Po-Shun Chuang
Biodiversity Research Center Academia Sinica
Taipei
Taiwan

Re: Spectrum00257-23R2 (Bacterial community shifts during polyp bail-out induction in *Pocillopora* corals)

Dear Dr. Po-Shun Chuang:

Thank you for thoroughly addressing the reviewers comments.

Your manuscript has been accepted, and I am forwarding it to the ASM Journals Department for publication. You will be notified when your proofs are ready to be viewed.

Sincerely,

Nikki Traylor-Knowles
Editor, Microbiology Spectrum
